# Assimilation of the AMSU-A radiances using the CESM (v2.1.0) and the DART (v9.11.13)/RTTOV CE1 (v12.3)

**Young-Chan Noh[1], Yonghan Choi[1], Hyo-Jong Song[2], Kevin Raeder[3], Joo-Hong Kim[1], and Youngchae Kwon[2]**

[1]Korea Polar Research Institute, Incheon, 21990, South Korea
[2]Department of Environmental Engineering and Energy, Myongji University, Seoul, 17058, South Korea
[3]National Center for Atmospheric Research, CISL/DAReS, Boulder, CO 80305, USA

**Correspondence:** Yonghan Choi (yhdchoi@kopri.re.kr)

**Abstract.** TS1 To improve the initial condition ("analysis") for numerical weather prediction, we attempt to assimilate observations from the Advanced Microwave Sounding Unit-A (AMSU-A) on board the low-Earth-orbiting satellites. The data assimilation system, used in this study, consists of the Data Assimilation Research Testbed (DART) and the Community Earth System Model as the global forecast model. Based on the ensemble Kalman filter scheme, DART supports the radiative transfer model that is used to simulate the satellite radiances from the model state. To make the AMSU-A data available to be assimilated in DART, preprocessing modules are developed, which consist of quality control, spatial thinning, and bias correction processes. In the quality control, two sub-processes are included, outlier test and channel selection, depending on the cloud condition and surface type. The bias correction process is divided into scan-bias correction and air-mass-bias correction. Like input data used in DART, the observation errors are also estimated for the AMSU-A channels. In the trial experiments, a positive analysis impact is obtained by assimilating the AMSU-A observations on top of the DART data assimilation system that already makes use of the conventional measurements. In particular, the analysis errors are significantly reduced in the whole troposphere and lower stratosphere over the Northern Hemisphere. Overall, this study demonstrates a positive impact on the analysis when the AMSU-A observations are assimilated in the DART assimilation system.

## 1 Introduction

Data assimilation is a numerical procedure for making the initial condition ("analysis") that is used as the starting point for a numerical weather prediction (NWP). In the data assimilation process, various observation data are combined with the short-term forecast ("background") derived from the NWP model, based on the error characteristics of the observations and model forecast (Kalnay, 2003). With the huge number of satellite observations and advances in model configurations (e.g., horizontal/vertical resolution and dynamic core) and data assimilation, the quality of the initial condition is significantly increasing, which enhances the forecast skill. In particular, the initial condition has dramatically improved since the satellite observations started to be assimilated (Migliorini et al., 2008; Eyre et al., 2020, 2022). This is because the satellites cover the regions where the conventional observations are sparse or absent. Among many types of satellite observations being assimilated, a significant forecast benefit mainly comes from the observations of the hyperspectral infrared and microwave sounders that provide unique information on the vertical structure of key atmospheric parameters (e.g., temperature and moisture) (Joo et al., 2013; Eresmaa et al., 2017; Menzel et al., 2018). For this reason, satellite observations are actively being assimilated into the data assimilation system in many operational NWP centers.

To advance the research related to data assimilation, a well-organized data assimilation system is essential, which consists of the forecast model, a data assimilation scheme, and flexible interfaces to use various types of observations. Operational NWP centers have well-constructed assimilation

systems to use diverse types of available observations with up-to-date data assimilation schemes. However, as most operational global NWP systems require huge computation resources, it is practically impossible for researchers to recreate those systems outside of the NWP centers. Thus, a user-friendly global data assimilation system is needed for small numerical modeling communities to attempt challenging studies related to advancing the data assimilation quality.

The National Center for Atmospheric Research (NCAR) has developed an open-source data assimilation tool that is named the Data Assimilation Research Testbed (DART) for data assimilation research, development, and education (Anderson et al., 2009). DART has interfaces to diverse Earth system components (e.g., atmosphere, ocean, and cryosphere) developed by many modeling centers. For instance, the Community Atmospheric Model (CAM), the atmospheric component of the Community Earth System Model (CESM) developed by NCAR, can be used to provide the short-range forecast that is the background field in DART. DART is based on the ensemble data assimilation method instead of the variational method, which requires complicated software specific to a particular numerical prediction model (Anderson et al., 2009; Raeder et al., 2012). In addition, well-defined modules are included to make various types of observations available in the DART data assimilation process. Thus, DART can assimilate many observation types (e.g., conventional and satellite-based wind). Liu et al. (2012) investigated the impact of the Global Positioning System (GPS) radio occultation (RO) observations on the forecast of Hurricane Ernesto (2006) using the DART assimilation system. Coniglio et al. (2019) showed that additional forecast benefit is made by assimilating the measurements of ground-based wind profilers. In addition, a decade-long reanalysis was created with 80 ensemble members derived from DART, using ground-based data, satellite-based winds, GPS-RO observations, and temperature soundings retrieved from the Atmospheric Infrared Sounder (AIRS) observation (Raeder et al., 2021).

However, there are few studies of assimilating satellite-measured radiances in the DART data assimilation system because the previous version of DART did not have the essential components, e.g., the radiative transfer model (RTM), needed to simulate the satellite radiances from the model state. Fortunately, in the recent version of DART (version 9.11.13), the RTM is included. The Radiative Transfer for TIROS Operational Vertical Sounder (RTTOV) version 12.3 is supported to map the model space into observation space in the data assimilation scheme (Saunders et al., 2018). In Zhou et al. (2022), the visible imagery of the Chinese geosynchronous-orbiting (GEO) satellite was assimilated in DART but using the Observing System Simulation Experiment (OSSE) framework in which the visible imagery was simulated and then assimilated. Considering that, it is interesting to assimilate the satellite-observed radiances using the DART data assimilation system to know how the analysis derived from DART is affected by real satellite observations.

Considering the fact that the analysis/CE2 forecast impact derived from the satellite radiances mainly comes from observations of hyperspectral infrared and microwave sounders (English et al., 2013; Joo et al., 2013; Kim and Kim, 2019), it is reasonable to assimilate the observations of both sounders first. Unfortunately, the use of hyperspectral infrared sounder observations was not supported in the recent version of DART. For this reason, we attempt to assimilate the radiances of the Advanced Microwave Sounding Unit-A (AMSU-A) temperature sounder within the DART data assimilation system coupled with the NCAR CESM. AMSU-A instruments are currently operating on board many low-Earth-orbiting (LEO) satellite platforms, and thus a large amount of AMSU-A observation data is available for assimilation. In addition, as the microwave sounder observations are less sensitive to clouds than the infrared sounder observations, the data availability of AMSU-A is better than that of the infrared sounder. AMSU-A observations are actively used to improve global/regional forecasts as well as severe weather forecasts such as tropical cyclones (Zhang et al., 2013; Zhu et al., 2016; Migliorini and Candy, 2019; Duncan et al., 2022). As the preprocessing modules (e.g., quality control, cloud detection, and spatial thinning) for AMSU-A observations are not provided in the DART package, they are developed in this study. In addition, the diagonal observation error covariance matrix is estimated using the method suggested by Desroziers et al. (2005), and a bias correction scheme is also developed based on the methods suggested by Harris and Kelly (2001). In this study, we attempt to assimilate the AMSU-A radiances in clear-sky conditions. In many operational NWP centers, the AMSU-A radiances have been assimilated in all-sky conditions (i.e., clear-sky and cloudy-sky) (Zhu et al., 2016; Migliorini and Candy, 2019; Duncan et al., 2022). However, as the current version of DART is not ready to assimilate the AMSU-A radiances in cloudy-sky conditions, only the clear-sky assimilation of AMSU-A radiances is considered. To assess the impact of assimilating AMSU-A observations on the analysis derived from DART, the assimilation experiments are conducted using the DART assimilation system coupled with the CESM as the forecast model system.

This paper is organized as follows. Section 2 provides the background information on the DART data assimilation system and CESM. Observation data assimilated in DART are described in Sect. 3. The developed preprocessing steps and the estimated observation errors are presented in Sects. 4 and 5, respectively. The setup of the assimilation experiments is explained in Sect. 6. The results of the first-guess/analysis CE3 departure analysis and the analysis impact are explored in Sect. 7, followed by a summary in Sect. 8.

## 2 DART–CESM data assimilation system

### 2.1 Data Assimilation Research Testbed (DART)

DART is an open-source assimilation package that has been developed by NCAR since 2002 for data assimilation development, research, and education. DART can be coupled with full-complexity Earth system components due to the flexible interfaces provided. In addition, the DART package provides the modules to convert observation data from a variety of native formats, e.g., the Binary Universal Form for the Representation of meteorological data (BUFR) format and the Hierarchical Data Format (HDF), into the input format specified for the DART system (Anderson et al., 2009; Raeder et al., 2012). The recent version of DART (version 9.11.13) is capable of using the RTTOV, a fast RTM, for assimilating visible, infrared, and microwave satellite observations. Provided in the RTTOV, many satellite instruments on board the GEO and LEO satellites are also supported in the DART assimilation package, but the hyperspectral infrared sounders, e.g., the Cross-track Infrared Sounder (CrIS) and the Infrared Atmospheric Sounding Interferometer (IASI), are excluded (Hoar et al., 2020). The main data assimilation technique provided by DART is the ensemble Kalman filter (EnKF) in which the forecast error covariance is estimated using short-range ensemble forecasts. The derived forecast error covariance is fully multivariate and depends on the synoptic situation.

### 2.2 Community Earth System Model (CESM)

CESM version 2 (CESM v2.1.0) is used as the model component of the ensemble data assimilation system. CESM2 is the latest generation of a coupled climate–Earth modeling system developed by NCAR, consisting of the atmosphere, land surface, ocean, sea-ice, land-ice, river, and wave models. These component models can be coupled to exchange states and fluxes (Hurrell et al., 2013; Kay at al., 2015). In this study, atmosphere and land component models are actively coupled, but the ocean component (sea surface temperature) and sea-ice coverage is specified by data read from files. As the atmosphere model of CESM2, CAM version 6 (CAM6) is an atmospheric general circulation model (AGCM) with the Finite Volume (FV) dynamical core (Danabasoglu et al., 2020). CAM6 provides the short-term forecast (6 h forecast) of the atmospheric state, which is used as the background state in the DART assimilation scheme. The land model is the Community Land Model version 5 (CLM5). The atmospheric variables are directly updated by the information derived from the observations ingested in the DART assimilation process, while the land state is affected interactively by the updated atmosphere state because the two component models are coupled. The two active models (CAM6 and CLM5) are run with a nominal 1° (1.25° in longitude and 0.95° in latitude) horizontal resolution. CAM6 has 32 vertical levels from the surface level to the top at 3.6 hPa (about 40 km).

## 3 Observations

### 3.1 NCEP PrepBUFR data

The baseline observation data are obtained from the National Centers for Environmental Prediction (NCEP) Automated Data Processing (ADP) global upper-air and surface weather observations that are available from the NCAR Research Data Archive (NCAR RDA) (https://rda.ucar.edu/datasets/ds337.0/ TS3). These data are produced in the PrepBUFR format for assimilation in the diverse NCEP NWP systems and mainly consist of ground-based observations and satellite-based wind retrievals. The ground-based observations include land and marine surface reports, aircraft reports, and radiosonde and pilot balloon (pibal) measurements, which are transmitted via the Global Telecommunications System (GTS) coordinated by the World Meteorological Organization (WMO). The satellite-based retrievals are provided by the National Environmental Satellite Data and Information Service (NESDIS). They include oceanic wind derived from the Special Sensor Microwave Imager (SSMI) and upper wind from the LEO and GEO satellites. As the NCEP ADP dataset is provided in the BUFR format, it must be converted to the data format available in the DART assimilation system, using the modules provided in the DART data assimilation package.

### 3.2 AMSU-A data

AMSU-A is the microwave temperature sounder that is currently on board diverse sun-synchronous satellite platforms e.g., MetOp satellites (MetOp-A, MetOp-B, and MetOp-C), three satellites of the National Oceanic and Atmospheric Administration (NOAA), and the National Aeronautics and Space Administration (NASA) research satellite Aqua. These three LEO satellite constellations provide near-global coverage, even in data assimilation that has a sub-daily assimilation window; NOAA satellites circle in an early-morning orbit (around 06:00 local time), MetOp satellites have a mid-morning orbit (around 09:00 local time), and Aqua has an afternoon orbit (around 13:00 local time). As a cross-track scanning sounder, the AMSU-A instrument has a total of 15 channels that consist of 12 channels (AMSU-A channels 3–14) over the 50–58 GHz oxygen ($O_2$) absorption band and three window channels (AMSU-A channels 1, 2, and 15) at 23.8, 31.4, and 89 GHz. The instrument measures 30 pixels in each swath with a spatial footprint size of 48 km in nadir. The channels over the $O_2$ absorption band mainly provide information about the vertical structure of tropospheric and stratospheric temperature (Mo, 1999; Goldberg et al., 2001). In this study, observations of AMSU-A instruments on board four LEO satellites (i.e., NOAA-19, Aqua, MetOp-A, and

MetOp-B) are assimilated within the DART data assimilation system.

## 4 Preprocessing AMSU-A observations

Prior to assimilating the AMSU-A observations into DART,
the AMSU-A observations should be passed through a preprocessing stage. Figure 1 shows the flowchart of the preprocessing stage for the AMSU-A observations as well as the DART assimilation step. In the preprocessing, three main steps are included: quality control, spatial thinning, and bias
correction. Quality control consists of two sub-processes, outlier test and channel selection, depending on the cloud condition and surface type. If the difference between the observed AMSU-A brightness temperature and the forward-modeled brightness temperature derived from the model
background (6 h forecast) is larger than 3 times the square root of the sum of the observation error variance and the prior background error variance, the AMSU-A observation is not assimilated (called outlier test). As the prior background error variance is based on the ensemble spread, the larger the
ensemble spread of the 6 h forecast, the more the AMSU-A observations are assimilated. More detailed information on the channel selection, spatial thinning, and bias correction process is described in Sect. 4.1, 4.2, and 4.3, respectively.

### 4.1 Channel selection for the cloud condition and
surface type

As each AMSU-A channel has distinct spectral characteristics, it is necessary to carefully choose the channels to be assimilated in the DART data assimilation system. First, the three AMSU-A channels at 23.8, 31.4, and 89 GHz
(i.e., channels 1, 2, and 15), distributed over the window region of the microwave spectrum, are not assimilated. These three window channels are mostly affected by the emitted radiances from the surface under clear-sky conditions, so there is almost no information about the
atmosphere. However, AMSU-A channels 1 (23.8 GHz) and 2 (31.4 GHz) are highly sensitive to clouds, so they are used for the quality control in which clouds are detected. In addition, even though the AMSU-A channels 3 (50.3 GHz) and 4 (52.8 GHz) are located over the $O_2$ absorption band
used for the temperature sounding, they have a strong sensitivity to the surface, so they are not used in DART. Considering that the upper parts of the weighting function of AMSU-A channels 12 ($57.29 \pm 0.322 \pm 0.022$ GHz), 13 ($57.29 \pm 0.322 \pm 0.010$ GHz), and 14
($57.29 \pm 0.3222 \pm 0.0045$ GHz) are above the top of the atmosphere (i.e., 3.6 hPa) in the CAM6, these three channels are also removed to prevent vertical interpolation errors that may occur in the forward modeling using the RTM. This leaves channels 5–11 ($53.596 \pm 0.115$, 54.4, 54.94, 55.5,

57.29, $57.29 \pm 0.217$, and $57.29 \pm 0.322 \pm 0.048$ GHz) as
the ones which may be assimilated.

As this study aims to assimilate the AMSU-A observations under clear-sky conditions, the cloud-affected channels are filtered out in the quality control step. In other words, the tropospheric channels (channels 5–7) whose peak of the weight-
55 ing function is below 200 hPa are rejected if the AMSU-A pixel is determined to be a cloud-affected pixel. To determine this, we calculate the cloud liquid water (CLW) derived from observations of AMSU-A channels 1 and 2 over the ocean, using the retrieval methodology suggested by Grody
et al. (2001). The CLW is defined as follows:

$$CLW = \cos\theta \, [D_0 + 0.754 \ln (285.0 - BT_{23})$$
$$- 2.265 \ln (285.0 - BT_{31})] \tag{1}$$
$$D_0 = 8.240 \, (2.622 - 1.846 \cos\theta) \cos\theta, \tag{2}$$

where $\theta$ is the satellite viewing zenith angle. $BT_{23}$ and $BT_{31}$
are the brightness temperature of AMSU-A channels 1 and 2, respectively. If the retrieved CLW is larger than 0.2 mm, the AMSU-A pixel is judged to be cloud-contaminated, and then the three tropospheric channels (channels 5–7) are rejected.

In this study, seven candidate AMSU-A channels (i.e.,
channels 5–11) are assimilated differently, depending on the surface type. Channels 5, 6, and 7 are the main tropospheric channels. Their weighting functions peak below 200 hPa but also have a bit of sensitivity to the surface because of the broad vertical shape of the weighting functions. Thus, the
quality of the analysis can be degraded by assimilating the three tropospheric channels over the land and sea-ice types whose surface information (e.g., surface temperature and surface spectral emissivity) is uncertain. For this reason, AMSU-A channels 5–7 are not assimilated over the land and
sea ice. To identify sea-ice area, the sea-ice index (SII) is retrieved from observations of AMSU-A channels 1 and 3 over the high-latitude region (poleward of 50°), using the retrieval algorithm suggested by Grody et al. (1999). The SII is derived as follows:

$$SII = 2.85 + 0.20 \, BT_{23} - 0.028 BT_{50}, \tag{3}$$

where $BT_{50}$ is the brightness temperature of AMSU-A channel 3. Three tropospheric channels are turned off if the SII is larger than 0.1 in the latitudes beyond 50°. However, as the surface information over the ocean is relatively reliable, seven candidate AMSU-A channels are assimilated under
90 clear-sky conditions. The AMSU-A channel list for DART is summarized in Table 1.

As an example, Fig. 2a and b present the spatial distribution of the CLW and the SII retrieved from AMSU-A on board NOAA-19 on 12 August 2014. It is found that many
regions over the ocean are covered by cloud-related systems (CLW > 0.2 mm) and also that sea ice (SII > 0.1) exists near the North and South Pole regions. Observations of AMSU-A channel 5 over the cloud region and sea-ice areas are re-

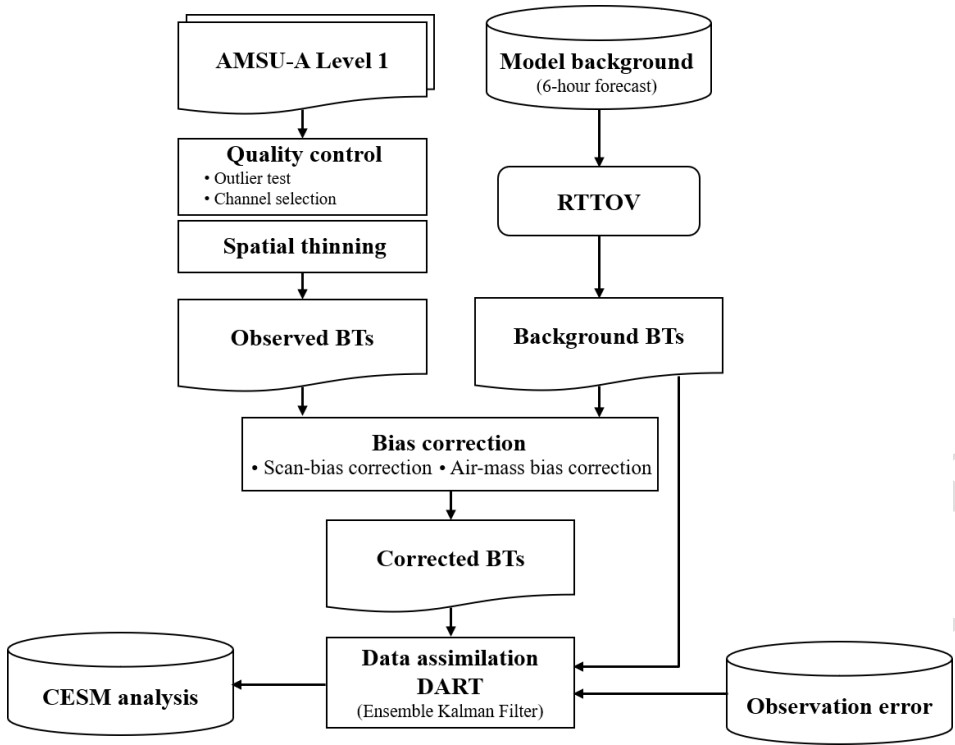

**Figure 1.** Flowchart of the preprocessing system for AMSU-A brightness temperatures (BTs).

**Table 1.** AMSU-A channel list for the DART data assimilation.

| Satellite platform | Type | CH5 | CH6 | CH7 | CH8 | CH9 | CH10 | CH11 |
|---|---|---|---|---|---|---|---|---|
| | Land/sea ice | | X | | O | O | O | O |
| Aqua | Ocean | NA* | O | NA | O | O | O | O |
| | Cloud | | X | | O | O | O | O |
| | Land/sea ice | X | X | X | | O | O | O |
| NOAA-19 | Ocean | O | O | O | NA | O | O | O |
| | Cloud | X | X | X | | O | O | O |
| | Land/sea ice | X | X | | | O | O | O |
| MetOp-A | Ocean | O | O | NA | NA | O | O | O |
| | Cloud | X | X | | | O | O | O |
| | Land/sea ice | X | X | X | O | O | O | O |
| MetOp-B | Ocean | O | O | O | O | O | O | O |
| | Cloud | X | X | X | O | O | O | O |

* NA: not available due to the malfunction in August and September 2014. O: assimilated. X: excluded.

jected (Fig. 2c). The channel selection process is also applied to the other two AMSU-A channels (channels 6 and 7), which are likely affected by clouds and sea ice. In the pretrial runs, it was found that the analysis quality is degraded if the AMSU-A observations are assimilated over Antarctica during the Southern Hemisphere winter season. This seems to be due to the complex topography of the Antarctic continent, extreme cold weather conditions, and large errors in the numerical model. Thus, AMSU-A observations are not used over the high-latitude region ($> 60°$ S) during the Southern Hemisphere winter season, in order to prevent the degradation of the analysis quality.

### 4.2 Spatial thinning

In addition to the inter-channel error correlation (refer to Sect. 5), spatial error correlation between the observations at a close distance also exists due to different representativeness

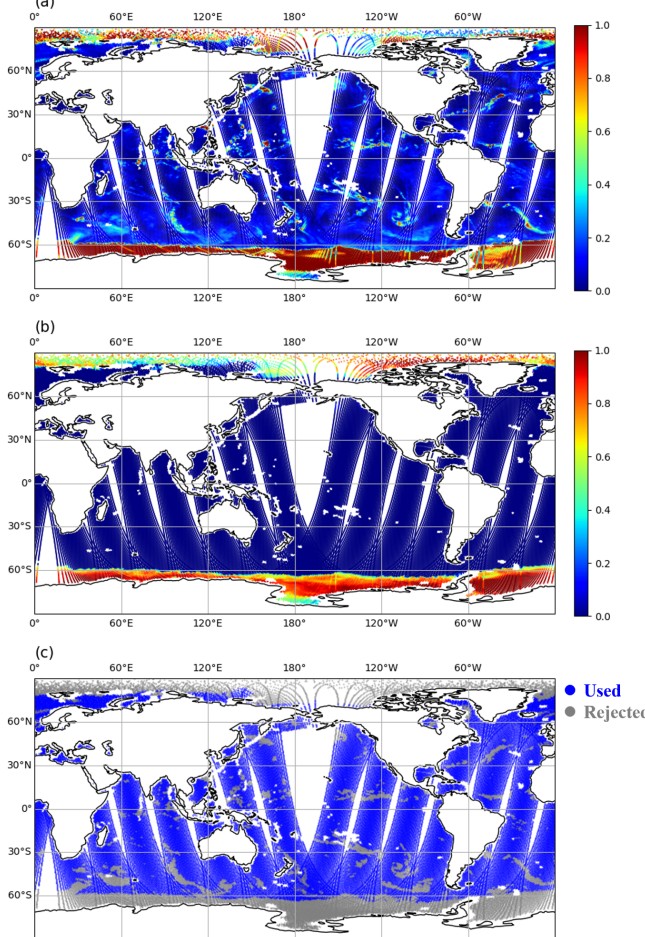

**Figure 2.** Spatial distribution of **(a)** cloud liquid water (CLW; mm), **(b)** sea-ice index (SII) retrieved from AMSU-A observations, and **(c)** quality flag of AMSU-A channel 5 (53.6 GHz) from NOAA-19 on 12 August 2014.

of the observed radiances and the model state, and the uncertain quality control process such as cloud detection (Ochotta et al., 2005; Bormann and Bauer, 2010). Thus, the analysis is likely to be sub-optimized if highly dense observations are assimilated without considering the spatial error correlations. A common treatment to counteract the spatial error correlation is spatial thinning, which is widely used in data assimilation systems operated by the NWP centers. To choose the optimal spatial thinning distance, we performed four extra assimilation runs in which different spatial thinning distance (i.e., 96, 192, 288, and 384 km) was applied. Except for the spatial thinning distance, these pre-trial runs were set up with the same assimilation factors, i.e., the estimated bias correction coefficients (refer to Sect. 4.3), the estimated observation errors (refer to Sect. 5), and the localization half-width of 0.075 (refer to Sect. 6). These distances are multiples of the AMSU-A field-of-view (FOV) footprint size (∼ 48 km in nadir). The thinning interval of 288 km resulted in the largest

analysis impact, so that distance was used to thin the observations in this study.

### 4.3 Bias correction

The biases mainly come from systematic errors: instrument calibration errors, inaccuracies of the RTM, and uncertain preprocessing (e.g., cloud detection errors). The biases tend to depend on the time of day and on the season as well as the instrument scan angle and air mass. While random errors are considered by defining the observation errors used in the assimilation process, the biases should be removed before assimilating the satellite observations. In these experiments, the biases are estimated using the time-averaged departures between the observed radiances and the simulated radiances from the spatiotemporally collocated model field (background) because of the absence of reference data suitable to compare the satellite observations (Scheck et al., 2018). The use of the simulated radiances from the model background (i.e., 6 h forecast) may be questionable because the model background could be biased. However, it is effectively impossible to find sufficient reference observations for comparing with these satellite observations, so the biases are made CE4 using the departures between the observed radiances and the model-simulated radiances. To estimate the systematic biases coming from diverse error sources, in this study, two bias correction processes are performed separately: scan-bias correction and air-mass-bias correction, using the statistical bias correction methods suggested by Harris and Kelly (2002).

As a cross-track microwave sounder, AMSU-A scans 30 FOVs per scan line, which are distributed symmetrically about the nadir. The scan angles of 30 FOVs range between ±48.33°. Thus, the observed radiance varies depending on the scan angle, even though the observation point is the same. The variation of AMSU-A radiance is due to the change in the optical path length between the Earth and the satellite instrument, called the limb effect. The variation of radiance along with the scan angle can be simulated in the RTTOV embedded in DART. However, the mean first-guess departures between the AMSU-A-observed radiances and forward-modeled radiances still increase with an increasing scan angle on the center of two near-nadir FOVs (15 and 16) (Fig. 3a), meaning that the residual scan-angle-dependent biases exist for each AMSU-A channel. Thus, the scan-bias correction is required to correct the residual scan bias for each AMSU-A channel. In this study, the scan-bias correction is performed using the pre-computed residual scan bias for each AMSU-A channel. There are two steps to estimate the residual scan bias for AMSU-A channels assimilated. First, the mean bias of the departure between the AMSU-A-observed radiances and forward-modeled radiances for each FOV is made with the data assimilation results derived from the pre-trial run. The pre-trial run was set up with the spatial thinning of 96 km (refer to Sect. 4.2) and the default localiza-

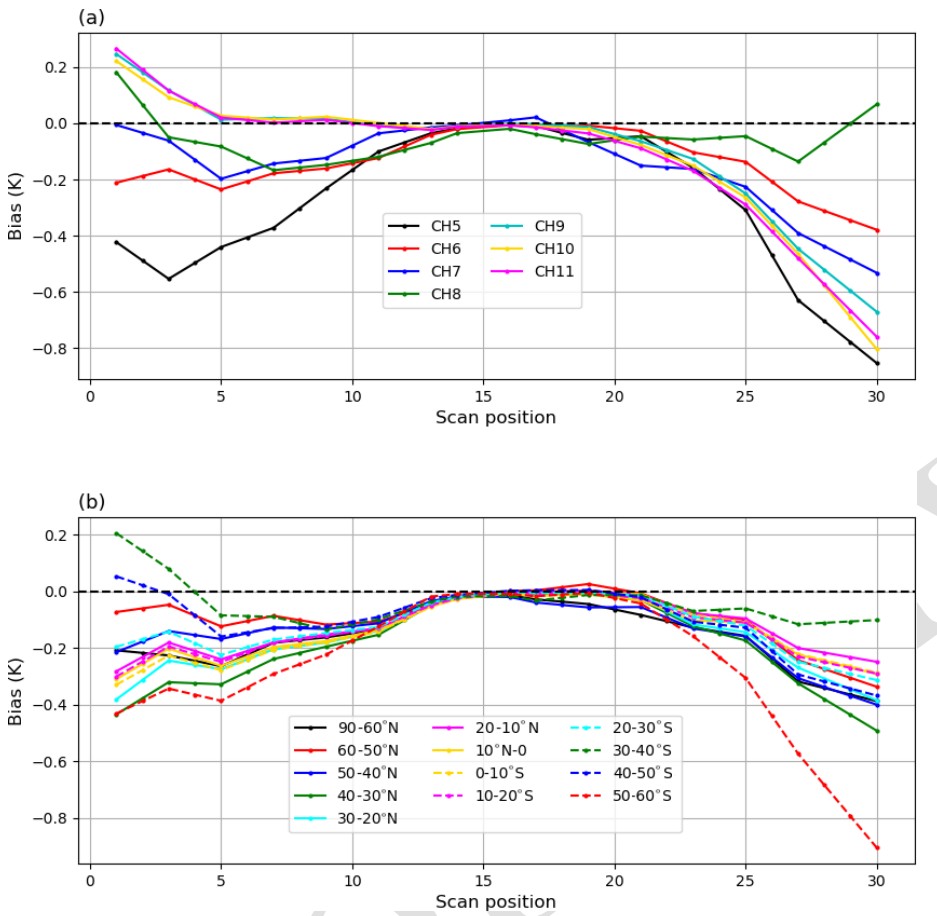

**Figure 3. (a)** Globally averaged, residual scan bias of AMSU-A channels 5–11 and **(b)** the regionally averaged, residual scan bias depending on 13 latitude bands for AMSU-A channel 6 on board MetOp-B during the period from 11 August to 25 August 2014.

tion half-width (0.15, refer to Sect. 6). The instrument noise errors were used as the observation errors within DART. Second, as the scan bias derived from the departures between the observed radiances and forward-modeled radiances likely includes the air-mass bias, the averaged residual scan bias is obtained by removing the mean bias of two near-nadir FOVs (15 and 16) from the bias for each FOV (1–30). In addition, as shown in Fig. 3b, it is also found that the residual scan biases have different patterns depending on the latitude band for AMSU-A channel 6 (not shown for other channels), suggesting that the use of globally averaged scan bias is likely to deteriorate the quality of AMSU-A data assimilation. Thus, the residual scan ($b^{\text{scan}}$) bias for each AMSU-A channel is subdivided into 14 latitude bands as follows:

$$b_i^{\text{scan}}(\theta, \phi) = \left[y - H(x_b)\right]_i (\theta, \phi) - \left[y - H(x_b)\right]_i$$
$$(\theta = 0, \phi), \tag{4}$$

where the subscript $i$ denotes the AMSU-A channel number ($i = 1, 2, \ldots, 15$), $\theta$ is the satellite scan angle, $\phi$ is the latitude band at an interval of $10°$ in the latitudes below 60 and $30°$ in the latitudes beyond $60°$, $y$ is the AMSU-A radiance, $x_b$ is the

background model state, and $H$ is the observation operator. Prior to the air-mass-bias correction, the observed brightness temperatures of each AMSU-A channel are corrected using the estimated scan-bias coefficients.

The air-mass bias ($b^{\text{airmass}}$) is predicted using the multivariate regression method. The biases are mainly due to uncertainties in the RTM, which tend to vary with the air-mass and surface characteristics. The predictors, used in the regression method, come from the model variables (i.e., 1000–300 hPa thickness, 200–50 hPa thickness, and surface temperature) that include information on air-mass and surface characteristics. The predictors regress to the first-guess departure between the satellite radiances and forward-modeled radiances as follows:

$$b_i^{\text{airmass}} = \beta_{i,0} + \sum_{j=1}^{N} \beta_{i,j} p_{i,j}, \tag{5}$$

where $\beta_{i,0}$ indicates the constant component of bias $b_i$, and $\beta_{i,j}$ denotes the bias correction coefficients of the predictor $p_{i,j}$. The subscripts $i$ and $j$ denote the AMSU-A channel

number and the predictor number (i.e., $j = 1, 2 \ldots N$), respectively.

For the tropospheric AMSU-A channels (channels 5–7), the air-mass bias is estimated with two model variables (i.e., 1000–300 hPa thickness and surface temperature) because the peak of the channel weighting function is positioned below the 200 hPa pressure level, and these channels have a bit of sensitivity to the surface. However, 200–50 hPa thickness is only employed for other upper-tropospheric and stratospheric AMSU-A channels (channels 8–11) whose peak of the weighting function is above 200 hPa. As the biases fluctuate with time, it is reasonable to update the regression coefficients and an intercept point periodically, rather than using the climatological-based coefficients that are estimated using the long-term model outputs. In this study, at each data assimilation cycle, the regression coefficients and an intercept point for each AMSU-A channel are computed using DART outputs for the last four cycles and then used to predict the air-mass biases. As shown in Fig. 4, the histograms of the first-guess departures of the MetOp-B channels 5–11 show a positive bias and a Gaussian distribution if the AMSU-A observations are not bias-corrected. In particular, channels 5 and 6 have a large positive bias of 1.0–1.5 K. However, the positive biases are almost removed through the bias correction process, meaning that the bias correction scheme works well (Table 2).

## 5  AMSU-A observation errors

As well as the model background error, the observation errors play an important role in determining the weight of the observations in the data assimilation system. Thus, it is an important step to define the observation errors so that the observations are suitably blended with the model background, which is a 6 h forecast derived from the CAM6, in order to provide the optimal initial condition to the numerical model. In this study, a diagonal observation error covariance matrix is used for the AMSU-A channels, meaning that the inter-channel error correlation is not considered. In fact, the use of the diagonal observation error covariance matrix may be problematic because the inter-channel error correlation definitely exists for the infrared and microwave sounders (Bormann and Bauer, 2010; Stewart et al., 2014; Weston et al., 2014; Campbell et al., 2017). Unfortunately, the recent version of DART (version 9.11.13) does not support the use of a full observation error covariance matrix in which the diagonal and off-diagonal components are fully defined. For this reason, the diagonal observation errors are empirically inflated to counteract the effect of error correlation between different AMSU-A channels. In other words, the inflated diagonal observation errors take account of the inter-channel error correlation as well.

To estimate the diagonal components (called variances) of the observation error covariance matrix TS5 ($\mathbf{R}$) for AMSU-

A channels, we use a diagnostic procedure suggested by Desroziers et al. (2005), in which the error variances are calculated with two departures, i.e., the background innovation (O-B CE5) between the observation ($y$) and the model background ($x_b$) and the analysis innovation (O-A CE6) between the observation and the model analysis ($x_a$), using the expression in Eq. (6).

$$\mathbf{R} = E\left[\{y - H(x_b)\}\{y - H(x_a)\}^T\right], \tag{6}$$

where $E$ is the statistical expectation operator, and the superscript "T" indicates the matrix transpose. To compute the observation error variances of AMSU-A channels on board four satellite platforms (i.e., Aqua, NOAA-19, MetOp-A, and MetOp-B), the background and analysis innovations were derived from the pre-trial run that was conducted from 25 August to 30 September 2014. In the pre-trial run, instrument noise errors were simply used as the observation errors. The pre-trial run was set up with the default localization half-width (0.15, refer to Sect. 6), the spatial thinning of 96 km (refer to Sect. 4.2), and the bias correction scheme (refer to Sect. 4.3). Then, the observation error variances were estimated using the Eq. (6).

As the surface-sensitive channels and upper-stratospheric channels are not assimilated in this study (see Sect. 4.1), Fig. 5 shows the observation errors of seven AMSU-A channels (channels 5–11) as well as the instrument noise errors employed in the pre-trial run. As some channels (i.e., channels 5 and 7 for Aqua, channel 8 for NOAA-19, and channels 7 and 8 for MetOp-A) malfunctioned during the trial period (11 August–30 September 2014), the errors for these channels were not needed or estimated. The estimated errors are larger than the instrument noise errors because various error sources (e.g., the radiative transfer modeling errors, representative errors, and systematic errors) are considered as well the instrument noise errors. The estimated errors for the tropospheric and upper-tropospheric channels (channels 5–9) are smaller than the errors for the stratospheric channels (channels 10–11). This error pattern is also presented for the instrument noise errors. As mentioned before, the estimated observation errors were inflated by a factor of 2 that was empirically estimated by the multiple pre-trial runs, in order to counteract the inter-channel error correlation. Then, the inflated observation errors, 2 times the estimated observation errors, were employed for the trial experiments, aiming at assessing the analysis impact of assimilating the AMSU-A observations.

## 6  Trial experiment design

To diagnose the analysis impact of assimilating the AMSU-A observations into the DART global data assimilation system, two assimilation experiments were conducted: (a) a control run (CNTL), where the conventional observations (i.e., ground-based observations and satellite-derived winds) were

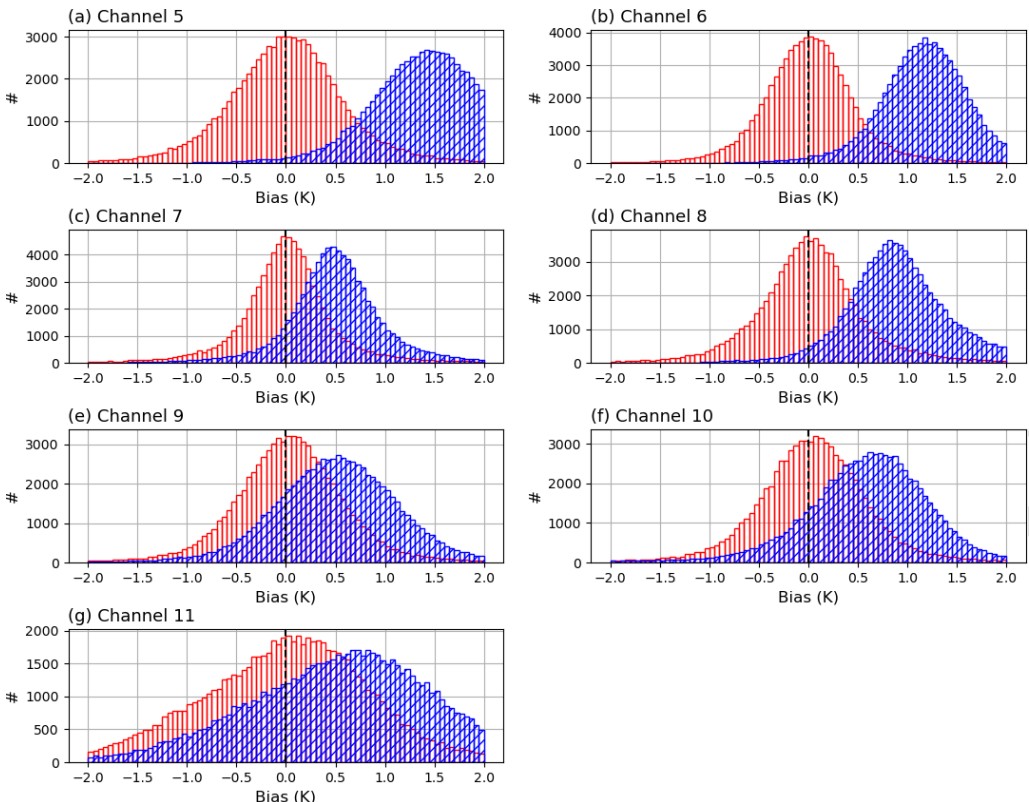

**Figure 4.** Histogram of the first-guess departures between the observations of the MetOp-B AMSU-A channels 5–11 and the corresponding model background (6 h forecast). Colors indicate the results before the bias correction (hatched blue) and after the bias correction (red), respectively.

**Table 2.** Mean biases and standard deviations of the first-guess departures (O-B) for MetOp-B AMSU-A channels before and after the bias correction.

| O-B | Bias correction | CH5 | CH6 | CH7 | CH8 | CH9 | CH10 | CH11 |
|---|---|---|---|---|---|---|---|---|
| Bias | X | 1.518 | 1.181 | 0.514 | 0.937 | 0.514 | 0.590 | 0.612 |
| | O | 0.0005 | 0.002 | 0.003 | 0.014 | 0.033 | 0.028 | 0.010 |
| SD TS4 | X | 0.677 | 0.489 | 0.521 | 0.572 | 0.639 | 0.688 | 1.052 |
| | O | 0.627 | 0.482 | 0.494 | 0.554 | 0.580 | 0.642 | 0.966 |

assimilated, and (b) the "AMSU-A run", where the AMSU-A observations from four LEO satellite platforms (i.e., Aqua, NOAA-19, MetOp-A, and MetOp-B) were assimilated as well as the conventional data that were assimilated in the CNTL run. For the AMSU-A run, the developed preprocessing steps (e.g., channel selection, thinning, and bias correction) were applied to the AMSU-A-observed radiances, and then the pre-computed AMSU-A observation errors were employed in the DART data assimilation process.

For two trial runs, available observation data were assimilated within a 6 h assimilation window from −3 to +3 h centered at the nominal analysis time (00:00, 06:00, 12:00, and 18:00 UTC). All trial runs were carried out four times a day for the trial period from 00:00 UTC 11 August to

18:00 UTC 30 September 2014. The CAM6 forecast model was run with a nominal 1° horizontal resolution (1.25° in longitude and 0.95° in latitude) and 32 vertical levels. The initial ensembles that are available at the NCAR RDA (https: //rda.ucar.edu/datasets/ds345.0/TS6) were obtained from the DART reanalysis. To adjust the effect of initial ensembles, a 2-week spin-up period (00:00 UTC 11 August to 18:00 UTC 24 August 2014) was included in the trial period. In this study, the ensemble adjustment Kalman filter (EAKF) is applied, which is a variation of the EnKF (Anderson, 2001). A total of 20 ensemble members were integrated to compute the flow-dependent background error covariance and the correlation between the DART state variables and observations.

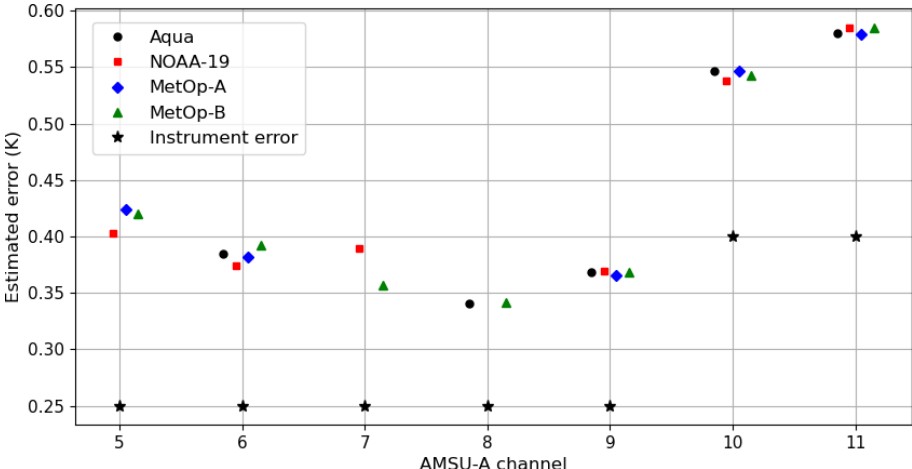

**Figure 5.** Estimated observation errors (K) for AMSU-A channels on board Aqua (black: circle), NOAA-19 (red: square), MetOp-A (blue: diamond), and MetOp-B (green: triangle) satellite platforms. Black asterisks indicate the instrument noise errors for AMSU-A channels.

All EnKF-based assimilation techniques have the sampling error that is induced by the limited size of the ensemble. In particular, the sampling error is likely to be large when the absolute value of correlation between the DART state variables and the observations is small. To remove the spurious correlation induced by limited ensemble size in DART, the correlation is multiplied by a localization factor that decreases from 1 to 0 with the physical distance between the model state variables and the observations. In DART, the localization half-width can be user-defined, which is half of the distance to where the localization factor is zero. To determine the localization half-width, three extra assimilation experiments were run with different half-widths (i.e., 0.15, 0.075, and 0.0375). Except for the localization half-width, the assimilation experiments were set up with the spatial thinning of 96 km (refer to Sect. 4.2), the bias correction scheme (refer to Sect. 4.3), and the estimated observation errors (refer to Sect. 5). As the largest analysis impact was made with the half-width of 0.075, the horizontal/vertical localization half-width of 0.075 rad was employed to prevent the use of erroneous correlation. However, as the model top height is much lower than the Earth's horizontal scale, the localization half-width in the vertical is normalized by the user-defined scale height, which is equivalent to 1 rad. In DART, the difference in scale height between the model top (360 Pa) and the standard surface pressure (101 325 Pa) is 5.73. In this study, the normalization scale height of 1.5, a default value in DART, was used, which is assumed to be equal to 1 rad. Thus, the localization half-width of 0.075 rad is converted into the scale height of 0.11, meaning that the localization cutoff can be an ellipsoid that is flat horizontally. In addition to the reduction of localization half-width (compared to the default value of 0.15), the sampling error correction algorithm was applied, which uses pre-defined information about the correlation between the model state variables and the observations

as a function of ensemble size. Detailed information on the sampling error correction algorithm is described in Anderson (2012).

The EnKF technique has a risk of underestimation of the ensemble spread, meaning that the ensemble estimates are too confident. If the ensemble spread becomes too small, the observation data are ignored in the data assimilation process, resulting in an ensemble collapse (Anderson et al., 2009; El Gharamti et al., 2019). To mitigate the underestimation issue of the ensemble spread, the uncertainty in the ensemble estimate is inflated by linearly moving each ensemble member away from the ensemble mean. It means that the standard deviation of the ensemble spread increases by applying the inflation value in a way that the ensemble mean is unchanged. In DART, the ensemble spread varies spatiotemporally, as a function of the evolving observation network and the chosen inflation algorithm. These experiments use a spatiotemporally varying inflation algorithm with a Gaussian distribution. More detailed information on the inflation algorithm adopted in DART is presented in El Gharamti et al. (2019).

## 7   Results

### 7.1   Assessment of first-guess departure and analysis departure

As the same conventional radiosonde measurements were assimilated in the two trial runs (i.e., CNTL and AMSU-A), the first-guess departure statistics between the radiosonde measurements and the spatiotemporally collocated background states (6 h forecast) can be used to assess the impact of the AMSU-A observations on the short-range forecast. Figure 6 shows the vertical structure of the standard deviation (SD) of the first-guess departure from the radiosonde temperature,

zonal wind, and meridional wind as well as the number of the radiosonde measurements used.

For the temperature, the first-guess departure errors are significantly reduced below 300 hPa for the AMSU-A runs as compared with the errors for the CNTL run (Fig. 6a). Because the AMSU-A channels provide vertical information about the air temperature, the temperature error reduction is the direct impact derived by assimilating the AMSU-A observations in the AMSU-A run. In addition to the radiosonde temperature, the first-guess departure errors decrease for the two wind components (i.e., zonal and meridional winds) (Fig. 6b and c). In particular, the SDs of the two winds at the 300 hPa level are reduced by up to about 4.7 m s$^{-1}$ TS7 in the AMSU-A run, compared to the error of about 5.1 m s$^{-1}$ for the CNTL run. As the model background error covariance includes the multivariate correlation between different model parameters (e.g., temperature and winds), a change in one model parameter can change another model parameter in the assimilation process. In addition, model parameters are linked in the governing equations and the physical parameterizations, which are embedded in the CAM6. That is, the change in one parameter results in the adjustment of another parameter in the model time integration. Thus, the error reduction of the wind components is the indirect impact of the improved temperature field by assimilating the AMSU-A observations.

In addition to the first-guess departure analysis of radiosonde, the assimilation impact of the AMSU-A observations can be diagnosed by comparing the first-guess departures of the AMSU-A with the analysis departures between the AMSU-A observations and the model analysis state. In general, if the observations are successfully assimilated, the SD of the analysis departure is smaller than that of the first-guess departure because the background fields are improved by assimilating the observations. As shown in Fig. 7, the SDs of the analysis departure are significantly smaller than those of the first-guess departure for AMSU-A-assimilated channels (channels 5–11), regardless of the satellite platforms, meaning that the AMSU-A observations have a positive analysis impact. In particular, the gap between the SDs of two departures is large for the stratospheric AMSU-A channels (channels 9–11).

## 7.2 Analysis impact of AMSU-A observations

To assess the impact of the AMSU-A observations on the analysis derived from the DART data assimilation system, the analysis errors are computed between the DART analysis and the European Centre for Medium-Range Weather Forecasts (ECMWF) reanalysis version 5 (ERA5) as the reference data. As the ERA5 is made through CE7 the assimilation of all available observation data in the ECMWF data assimilation system and provides consistent maps without spatial gaps, the ERA5 is employed to assess the model-derived output. For four primary atmospheric parameters (i.e., 500 hPa geopotential height, temperature, zonal wind, and meridional wind), the departures between the DART ensemble-mean analysis and the ERA5 are computed. Then bias and standard deviation are derived from the long-term departures. In particular, the error of 500 hPa geopotential height is widely used to assess the overall performance of the model-derived output because large-scale atmospheric motion in the middle troposphere (500 hPa) is closely linked with lower-level atmospheric motion.

Figure 8 describes the mean bias and SD of 500 hPa geopotential height for the CNTL and AMSU-A run, depending on the latitudinal regions. Detailed error values are described in Table 3. For two trial runs, overall negative mean bias occurs, reaching up to about −18 m. However, the bias difference varies depending on the latitudinal regions. Over the Northern Hemisphere (30–90° N), the AMSU-A run has a larger negative bias than the bias for the CNTL run. However, over the tropics (30° S–30° N) and Southern Hemisphere (30–90° S), the CNTL run has a larger negative bias than the bias for the AMSU-A run. Thus, similar global mean bias (about −18 m) for two trial runs is caused by the offsetting between regionally different bias patterns.

Considering that the geopotential height is a primary function of the average air temperature between the surface and the pressure level, we assumed that the model temperature has a cold bias at least below the 500 hPa pressure level. As expected, it is found that a negative bias is presented in the temperature field for both two trial runs (not shown). In addition, as shown in Fig. 9, the first-guess departure of the radiosonde temperature for the two trial runs has large positive values, implying that a cold bias exists in the model temperature fields (6 h forecast). In Raeder et al. (2021), it was noted that the CAM6/DART-derived reanalysis has a cold bias in the troposphere. However, it is still unclear as to why the CAM6-based temperature fields have a cold bias. The bias issue in CAM6 will be an interesting study in future work.

Even though the AMSU-A observations, including the temperature information, are additionally assimilated in the AMSU-A run, the AMSU-A run has a negative temperature bias that occurs in the CNTL run. It is related to the bias correction applied to the AMSU-A observations in DART. As mentioned in Sect. 4.3, the AMSU-A radiances are corrected by eliminating the biases based on the departure between the observed radiances and the forward-simulated radiances from the model background field. In addition, in this study, the bias correction coefficients were even updated at each cycle, using the DART outputs from the last four cycles. Thus, the information on the model bias is included in the biases derived from the correction scheme, which gradually fits the observations to the model background over the sequent assimilation cycles. As a result, the model bias still exists in the AMSU-A run as well as the CNTL run.

However, the global-mean SD of 500 hPa geopotential height for the AMSU-A run is reduced to about 42 m as compared with the SD (about 49 m) for the CNTL run, meaning

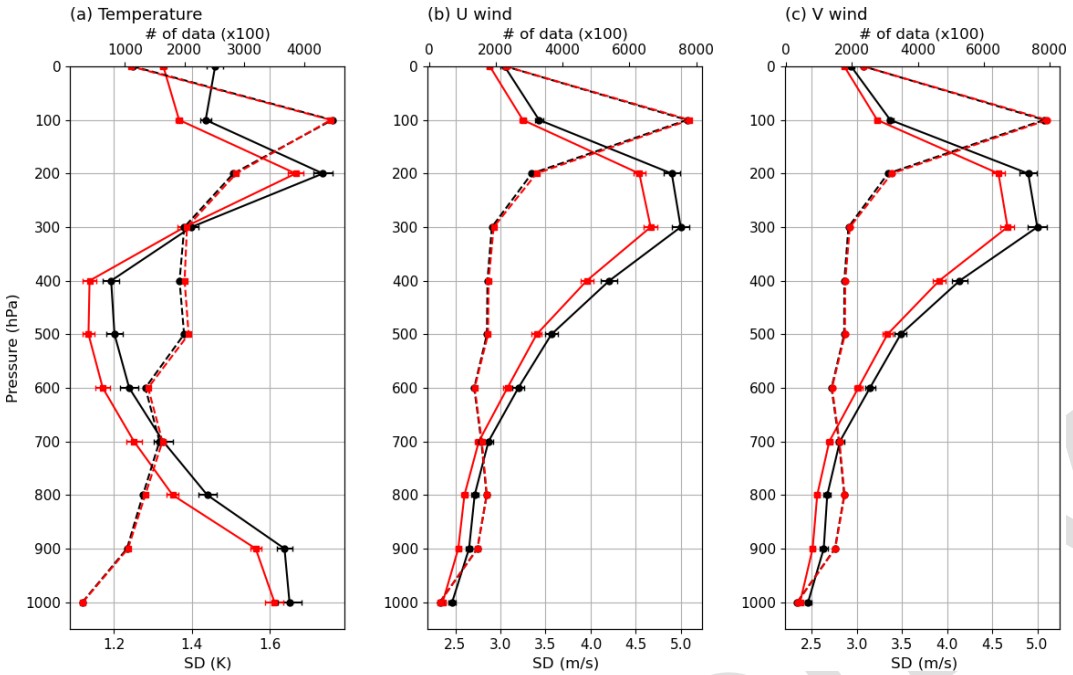

**Figure 6.** The standard deviation (SD) of the first-guess departures for the radiosonde **(a)** temperature, **(b)** zonal wind, and **(c)** meridional wind for the control (CNTL run: circle symbol and black line) and experiment (AMSU-A run: square symbol and red line) runs. Solid and dashed lines indicate the SD and the number (top axis) of radiosonde measurements assimilated, respectively. The 99 % confidence intervals are indicated by the horizontal black lines.

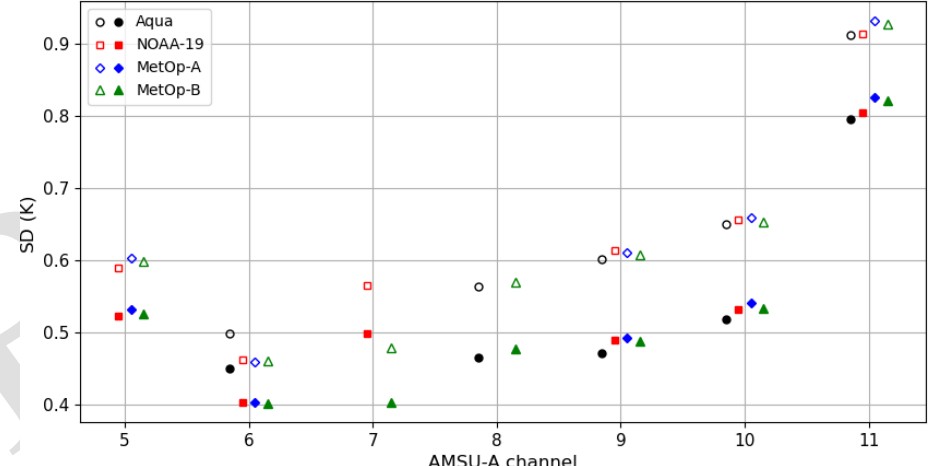

**Figure 7.** The standard deviations (SDs) of the first-guess departure (unfilled symbols) and analysis departure (filled symbols) for AMSU-A channels on board Aqua (black: circle), NOAA-19 (red: square), MetOp-A (blue: diamond), and MetOp-B (green: triangle) satellites.

that the 500 hPa geopotential height predictions are improved by assimilating the AMSU-A observations (Table 3). In particular, the error is largely reduced over the Northern Hemisphere. That is, the analysis impact is more significant in the Northern Hemisphere. It is inconsistent with the consensus that the assimilation impact of satellite observations is larger in the Southern Hemisphere, where the conventional data are sparse (Terasaki and Miyoshi, 2017; Yamazaki et al., 2023).

As shown in Fig. 10a and b, a positive impact mainly occurs in the high-latitude region ($> 60°$ N). In contrast, over the tropics and Southern Hemisphere, the error reduction is relatively smaller than over the Northern Hemisphere. In the tropics, the analysis error (about 14 m) is quite small for the CNTL run, as compared with the large errors of about 48 and 63 m in the Northern Hemisphere and Southern Hemisphere, respectively. Following Judt (2020), it was demon-

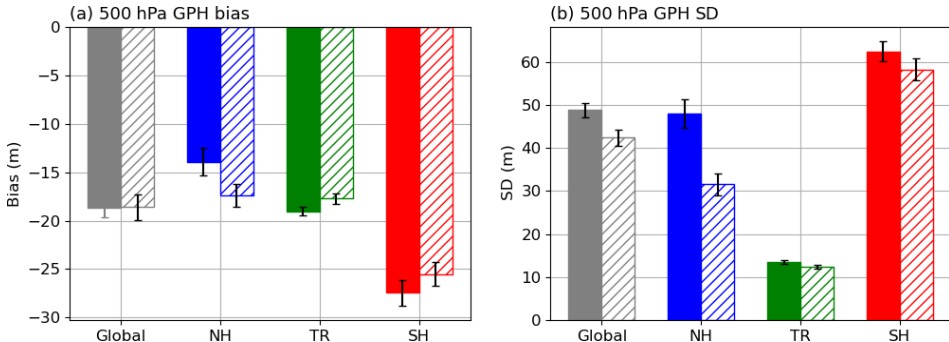

**Figure 8. (a)** Mean bias and **(b)** standard deviation (SD) of the analysis of 500 hPa geopotential height over the global (grey), Northern Hemisphere (NH: blue), tropics (TR: green), and Southern Hemisphere (SH: red), derived against the ERA5 reanalysis. Filled and hatched bars indicate the results for the control (CNTL) and experiment (AMSU-A) run, respectively. The 99 % confidence intervals are indicated by the vertical black lines.

**Table 3.** Error statistics of 500 hPa geopotential height (m) for the control (CNTL run) and experiment (AMSU-A run) run. Better values are bolded. In parentheses, error statistics are shown over the midlatitude region (30–60° S and 30–60° N) in the Northern and Southern Hemisphere.

| Trial name | Bias | | | | SD | | | |
|---|---|---|---|---|---|---|---|---|
| | Global | NH | TR | SH | Global | NH | TR | SH |
| CNTL | −18.70 | −13.90 (−18.43) | −19.05 | −27.45 (−19.84) | 48.82 | 48.02 (26.71) | 13.55 | 62.54 (38.55) |
| AMSU-A | −18.59 | −17.39 (−16.95) | −17.73 | −25.51 (−19.54) | 42.42 | 31.55 (20.24) | 12.41 | 58.29 (33.49) |

strated that the tropical atmosphere has longer predictability than the extratropical atmosphere. Thus, the AMSU-A observations are conservatively assimilated in the tropics due to the small forecast errors, leading to less analysis impact.

It is noted that the AMSU-A assimilation impact is neutral in the high-latitude region (> 60° S) over the Southern Hemisphere. In contrast, in the high-latitude region (> 60° N) over the Northern Hemisphere, the assimilation impact is significant. It is because the AMSU-A observations were not assimilated in the high-latitude region (> 60° S) over the Southern Hemisphere during the Southern Hemisphere winter season when the trial runs were conducted (mentioned in Sect. 4.1), resulting in the neutral analysis impact. Thus, if the high-latitude regions (i.e., 60–90° S and 60–90° N) are extracted in the error computation over both hemispheres, the analysis impact is still significant, but the difference in the analysis impact between both hemispheres considerably decreases (Table 3). It is still a challenging issue to assimilate the satellite radiances over the Antarctic continent because of the complex topography, extreme weather condition, and large errors in the numerical model. In particular, as the conventional observations are quite sparse in the high-latitude region, the forecast errors are relatively larger than the other latitudinal regions (i.e., the tropics and midlatitude region, shown in Fig. 10a). In addition, the trial period (11 August–

30 September 2014) is the Southern Hemisphere winter season when the Antarctic continent was under extremely cold weather conditions. In fact, in the pre-trial run, we found that the analysis field was degraded near the Antarctic continent by assimilating the AMSU-A observations. Thus, to prevent the analysis degradation, the AMSU-A observations were rejected over the high-latitude region (> 60° S) in the Southern Hemisphere. The assimilation of the AMSU-A observation in the Antarctic region will be handled in future work.

Figure 11 shows the normalized difference of SD of temperature, zonal wind, and meridional wind between the AMSU-A run and CNTL run, depending on the latitudinal regions (i.e., global, Northern and Southern Hemisphere, and tropics). The SD difference is normalized by the SD for the CNTL run. A negative value means that assimilating the AMSU-A observations provide analysis benefit. In contrast, a positive value indicates that the analysis error increases for the AMSU-A run compared with the error for the CNTL run, implying a negative analysis impact of the AMSU-A observations.

For the temperature, the global-mean analysis errors are significantly reduced in the whole troposphere and lower stratosphere for the AMSU-A run, as compared with the CNTL run. Large error reduction occurs in the lower stratosphere (−28 % and −21 % in 100 and 200 hPa, respectively),

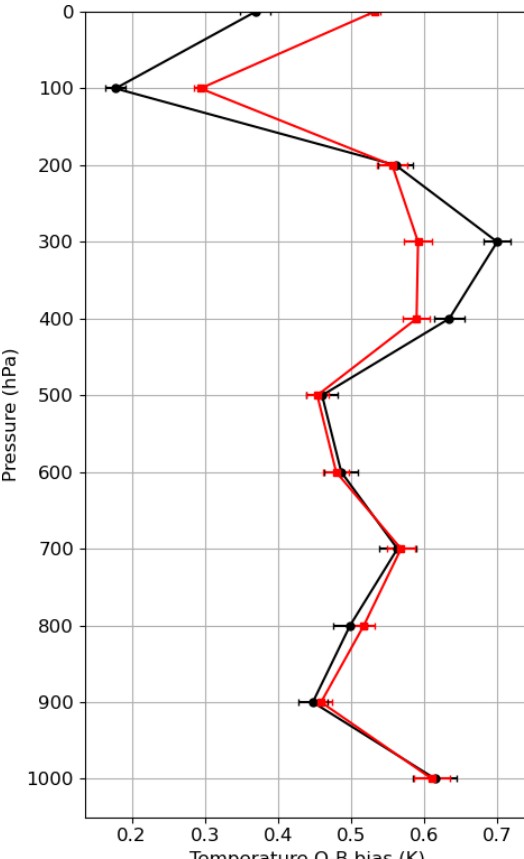

**Figure 9.** Mean bias of the first-guess departure for the radiosonde temperature measurements for the control (CNTL run: circle symbol and black line) and experiment (AMSU-A run: square symbol and red line) runs. Horizontal lines indicate 99 % confidence intervals.

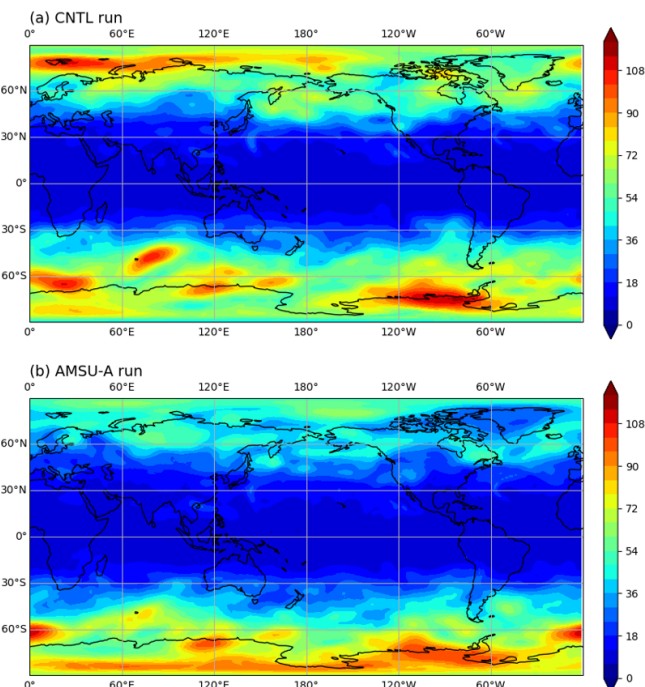

**Figure 10.** Spatial distribution of the standard deviation (SD) of the analysis of 500 hPa geopotential height for the **(a)** control run (CNTL) and **(b)** experiment (AMSU-A) runs, derived against the ERA5 reanalysis.

which is consistent with the large gap between the SDs of the first-guess departure and the analysis departure for the stratospheric AMSU-A channels (channels 9–11) whose peak of the weighting function is above 200 hPa (shown in Fig. 7). Similar to the results of the 500 hPa geopotential height, a strong error reduction mainly occurs in the Northern Hemisphere where the error reduces up to about 28 % in the 500 hPa pressure level (Fig. 11a). The error decrease trends are consistent with the trends of the first-guess departure errors of the radiosonde temperature measurements in which a significant error decrease occurs in the 500 hPa layer (Fig. 6a). However, in the lower stratosphere (100 hPa pressure level), the analysis error decreases up to about 45 % in the Southern Hemisphere.

For two wind components (i.e., zonal and meridional winds), similar to the results of the temperature, the global-mean analysis errors for the AMSU-A run overall decrease in the whole troposphere and lower stratosphere (Fig. 11b and c). It is noted that the magnitude of the error decrease tends to increase with height, reaching about −13 % in the

100 hPa level for the zonal and meridional wind. Moreover, most analysis impact is made in the Northern Hemisphere, except in the 100 hPa level, where the maximum error decrease occurs in the Southern Hemisphere. However, over the Southern Hemisphere, the analysis errors for the AMSU-A runs are larger than the errors for the CNTL run in the middle and lower troposphere. For the spatial pattern of the SD of two wind components (not shown), it is found that the error increment mainly occurs in the high-latitude region (> 60° S), where the AMSU-A data were not assimilated for the AMSU-A run. Considering that the temperature fields above the latitude of 60° S were only updated by the AMSU-A assimilation, the analysis degradation is possibly due to the discontinuity of the latitudinal temperature gradient near the latitude of 60° S.

In the model humidity field, a positive analysis impact only occurs in the Northern Hemisphere (not shown) but is not as significant as the abovementioned parameters (i.e., 500 hPa geopotential height, temperature, and winds). As a further study, we plan to assimilate the Microwave Humidity Sounder (MHS), providing information on the vertical structure of humidity so that the initial condition of model humidity is improved.

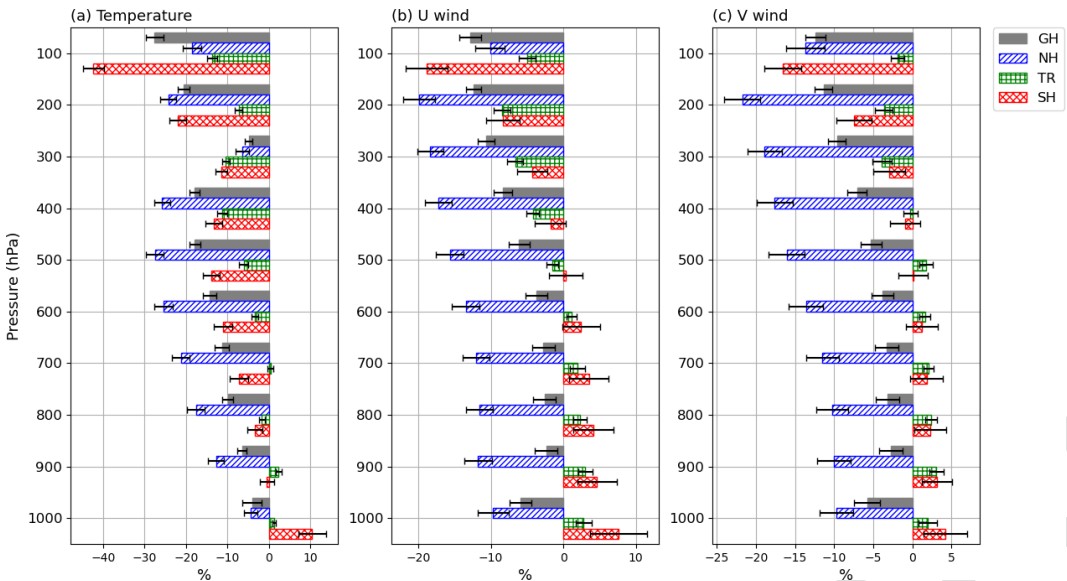

**Figure 11.** Normalized difference of the standard deviation (SD) of the analysis of **(a)** temperature, **(b)** zonal wind, and **(c)** meridional wind between the experiment (AMSU-A) run and the control (CNTL) run, derived against the ERA5 reanalysis. Hatched colors indicate the latitude regions (global: grey, Northern Hemisphere: blue, tropics: green, and Southern Hemisphere: red). Horizontal lines indicate 99 % confidence intervals.

# 8  Summary

In this study, we attempted to assimilate the AMSU-A observations using the global data assimilation system consisting of DART and CESM. To make the AMSU-A data available to be assimilated, preprocessing steps were developed, which include quality control (i.e., outlier test and channel selection), spatial thinning, and bias correction (i.e., scanbias correction and air-mass-bias correction). In addition, the observation error covariance matrix was estimated, but only its diagonal components were employed in DART because the inter-channel error correlation is not considered in the current version of DART. To counteract the inter-channel error correlation, the diagonal components were inflated.

To assess the impact of the AMSU-A observations on the DART-derived analysis, trial experiments were conducted from 11 August to 30 September 2014. The derived analysis fields were verified using the ERA5 as the reference. For the primary atmospheric parameters (i.e., 500 hPa geopotential height, temperature, zonal wind, and meridional wind), an additional analysis benefit is provided by assimilating the AMSU-A observations on top of the DART data assimilation system which already makes use of the conventional ground-based observations. In particular, a large analysis impact is shown in the Northern Hemisphere, where the analysis errors of the temperature and two wind components are significantly reduced in the whole troposphere. However, in the tropics, the analysis impact is relatively small due to the small forecast errors. Compared with the Northern Hemisphere, less analysis impact in the Southern Hemi-

sphere seems to be due to the reduction in the number of assimilated AMSU-A observations. The AMSU-A observations are rejected in the high-latitude regions ($> 60°$ S) during the Southern Hemisphere winter season because assimilating these observations worsens the analysis quality.

*Code and data availability*. DART version 9.11.13 was obtained from https://github.com/NCAR/DART TS8. CESM version 2.1.0 is released at https://github.com/ESCOMP/CESM/tree/release-cesm2.1.0 TS9. Atmospheric initial conditions and the baseline observations at the BUFR format were obtained from the NCAR RDA (https://rda.ucar.edu/datasets/ds337.0 TS10 or https://doi.org/10.5065/Z83F-N512, National Centers for Environmental Prediction/National Weather Service/NOAA/U.S. Department of Commerce, 2008). AMSU-A Level-1B version 5 data from the Aqua satellite, including the calibrated brightness temperatures, were downloaded from the NASA Goddard Earth Sciences Data and Information Services Center (https://www.earthdata.nasa.gov/eosdis/daacs/gesdisc TS11). In addition, AMSU-A Level-1B from NOAA-19, MetOp-A, and MetOp-B satellites were downloaded from the atmosphere product section in the EUMETSAT product navigator (https://navigator.eumetsat.int TS12). The ECMWF ERA5 hourly data on pressure levels were acquired from the Copernicus Climate Change Service (C3S) Climate Data Store (https://cds.climate.copernicus.eu/cdsapp#!/dataset/reanalysis-era5-pressure-levels TS13). As well as the software codes, the model outputs are available at https://doi.org/10.5281/zenodo.7714755 (Noh, 2023a) and https://doi.org/10.5281/zenodo.7983459 (Noh, 2023b) TS14.

*Author contributions.* YCN and YC conceptualized the research idea. YCN and YC developed the methods with assistance from HJS and YK. YCN led the writing of the paper with support from YC, HJS, and KR. YC, HJS, KR, and JHK were involved in writing the final version of the paper, whereas YK provided feedback on it.

*Competing interests.* The contact author has declared that none of the authors has any competing interests.

*Acknowledgements.* This project is sponsored by a Korea Polar Research Institute (KOPRI) grant, funded by the Ministry of Oceans and Fisheries (KOPRI PE23010). NCAR is supported by the US National Science Foundation (NSF). Any opinions expressed here are not necessarily those of NCAR or the NSF. Hyo-Jong Song and Youngchae Kwon are supported by the Korea Environment Industry & Technology Institute (KEITI) through "Climate Change R&D Project for New Climate Regime", funded by the Ministry of Environment (MOE) of South Korea (2022003560006).

*Financial support.* This research has been supported by the NAME OF FUNDER (grant no. GRANT AGREEMENT NO). TS15

*Review statement.* This paper was edited by Yuefei Zeng and reviewed by Yongbo Zhou, Lukas Kugler, and Wei Han.

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

## Remarks from the language copy-editor

## Remarks from the typesetter

**TS8**    Please clarify whether the data set/code is your own. If yes, please provide a DOI in addition to your GitHub URL since our reference standard includes DOIs rather than URLs. If you have not yet created a DOI for your data set, please issue a Zenodo DOI (https://help.github.com/en/github/creating-cloning-and-archiving-repositories/referencing-and-citing-content). If the data set/code is not your own, please inform us accordingly. In any case, please ensure that you include a reference list entry corresponding to the data set including creators, title, and date of last access.

**TS9**    Please clarify whether the data set/code is your own. If yes, please provide a DOI in addition to your GitHub URL since our reference standard includes DOIs rather than URLs. If you have not yet created a DOI for your data set, please issue a Zenodo DOI (https://help.github.com/en/github/creating-cloning-and-archiving-repositories/referencing-and-citing-content). If the data set/code is not your own, please inform us accordingly. In any case, please ensure that you include a reference list entry corresponding to the data set including creators, title, and date of last access.