# Peer review of "Assimilation of the AMSU-A radiances using the CESM (v2.1.0) and"

_Geoscientific Model Development, 2023_

## Author Comment (AC1)

**Referee #1**

We sincerely appreciate your comments and concerns about this manuscript. Detailed responses to your comments are described below. In this response letter, to distinguish between the referee's comments and authors' responses, your original comments are presented in an Italic font with an underline.

*Review of "Assimilation of the AMSU-A radiances using the CESM (v2.1.0) and the DART (v9.11.13)/RTTOV (v12.3)" by Young-Chan Noh et al.*

*This paper develops and evaluates pre-processing modules for the assimilation of AMSU-A microwave observations from low-earth-orbiting satellites. The authors developed quality control and bias correction and validated their approach by data-denial experiments. They assess forecast impact by comparing the (6-h forecast) first-guess departures between experiments. Results show that the RMSE of 6-h forecasts is reduced compared to radiosonde observations and ERA5 as reference.*

*I want to thank the authors for their valuable contribution as it allows the assimilation of AMSU-A observations for reanalysis and weather forecast. I found this manuscript very interesting, however, I think some issues need to be addressed before publication.*

*Major comments*

*Please write in the introduction how your work relates to previous research on microwave and/or AMSU-A radiance assimilation. Readers should be made aware of relevant previous microwave assimilation studies if available.*

[Reply]
Thank you for your comment. I agree with your opinion. We modify the introduction part as follows:
[Old, lines 76-79]
"For this reason, we attempt to assimilate the radiances of the Advanced Microwave Sounding Unit-A (AMSU-A) temperature sounder within the DART data assimilation system coupled with the NCAR CESM. AMSU-A instruments are currently operating on board many low-earth-orbiting (LEO) satellite platforms, and thus a large amount of AMSU-A observation data is available for assimilation. In addition, as the microwave sounder observations are less sensitive to clouds than the infrared sounder observations, the data availability of AMSU-A is better than that of the infrared sounder. As the preprocessing modules (e.g., quality control, cloud detection, and spatial thinning) for AMSU-A observations are not provided in the DART package, they are developed in this study. In addition, the diagonal observation error covariance matrix is estimated using the method suggested by Desroziers et al. (2005), and the bias correction scheme is also developed based on the methods suggested by Harris and Kelly (2001)."
[New]
"For this reason, we attempt to assimilate the radiances of the Advanced Microwave Sounding Unit-A (AMSU-A) temperature sounder within the DART data assimilation system coupled with the NCAR CESM. AMSU-A instruments are currently operating on board many low-earth-orbiting (LEO) satellite platforms, and thus a large amount of AMSU-A observation data is available for assimilation. In addition, as the microwave sounder observations are less sensitive to clouds than the infrared sounder observations, the data availability of AMSU-A is better than that of the infrared sounder. AMSU-A observations are actively used to improve global/regional forecasts as well as severe weather forecasts

such as tropical cyclones (Zhang et al., 2013; Zhu et al., 2016; Migliorini and Candy, 2019; Duncan et al., 2022). As the preprocessing modules (e.g., quality control, cloud detection, and spatial thinning) for AMSU-A observations are not provided in the DART package, they are developed in this study. In addition, the diagonal observation error covariance matrix is estimated using the method suggested by Desroziers et al. (2005), and a bias correction scheme is also developed based on the methods suggested by Harris and Kelly (2001). In this study, we attempt to assimilate the AMSU-A radiances in clear-sky conditions. In many operational NWP centers, the AMSU-A radiances have been assimilated in all-sky conditions (i.e., clear-sky and cloudy-sky) (Zhu et al., 2016; Migliorini and Candy, 2019; Duncan et al., 2022). However, as the current version of DART is not ready to assimilate the AMSU-A radiances in cloudy-sky conditions, only the clear-sky assimilation of AMSU-A radiances is considered."

Duncan, D. I., Bormann, N., Geer, A. J., and Weston, P.: Assimilation of AMSU-A in All-Sky Conditions, Mon. Weather Rev., 150, 1023-1041, doi: 10.1175/MWR-D-21-0273.1, 2022.

Migliorini, S. and Candy, B.: All-sky satellite data assimilation of microwave temperature sounding channels at the Met Office, Q. J. Roy. Meteor. Soc., 145, 867–883, https://doi.org/10.1002/qj.3470, 2019.

Zhang, M., Zupanski, M., Kim, M.-J., and Knaff, J. A.: Assimilating AMSU-A Radiances in the TC Core Area with NOAA Operational HWRF (2011) and a Hybrid Data Assimilation System: Danielle (2010), Mon. Weather Rev., 141, 3889–2907, https://doi.org/10.1175/MWR-D-12-00340.1, 2013.

Zhu, Y., Liu, E., Mahajan, R., Thomas, C., Groff, D., Delst, P. V., Collard, A., Kleist, D., Treadon, R., and Derber, J. C.: All-Sky Microwave Radiance Assimilation in NCEP's GSI Analysis System, Mon. Weather Rev., 144, 4709–4735, https://doi.org/10.1175/mwr-d-15-0445.1, 2016.

*If available, how do your results compare to results of other data-denial studies assimilating AMSU-A? Did you find any similar studies you can compare your results to? Are they similar or different in any ways? You could comment on that in the summary.*

[Reply]
The AMSU-A radiances are already operationally assimilated at the many operational NWP centers such as the ECMWF, Met Office, and KMA. In particular, the AMSU-A instrument is one of three instruments comprising the Advanced TIROS Operational Vertical Sounder (ATOVS). Other two instruments are the High Resolution Infrared Sounder (HIRS) and the Microwave Humidity Sounder (MHS). Most global NWP centers conducted data-denial studies in which the ATOVS data were assimilated or rejected, instead of solely assimilating the AMSU-A radiances (English et al., 2000; Bormann et al., 2008). Thus, it is difficult to compare the results of this study with the previous data-denial studies conducted by the operational NWP centers.

However, there are two previous studies related to the assimilation of AMSU-A radiances into the global NWP system. First, Terasaki and Miyoshi (2017) attempted to assimilate the AMSU-A radiances into the non-hydrostatic icosahedral atmospheric model (NICAM)-based local ensemble transform Kalman filter (LETKF), resulting in the significant improvement of the analysis field. Second, Yamazaki et al. (2023) assessed the assimilation impact of AMSU-A observations using two diagnostic techniques which are ensemble-based forecast sensitivity to observations (EFSO) and observing system experiments (OSEs). As only three AMSU-A channels (channels 6, 7, and 8) were assimilated, it might be unfair to compare Yamazaki's results with our results. In addition, in Yamazaki et al. (2023), a single integrated skill score, called the observation impact (OI), was only used to diagnose the assimilation impact of AMSU-A observations. Nevertheless, Yamazaki et al. (2023) demonstrated that assimilating

AMSU-A observations contribute to improving the analysis quality. Thus, considering the overall context, instead of mentioning Yamazaki's study in the summary, it seems to be better to mention it in the result section where the analysis impacts are mentioned. We revised these sentences as follow:

[Old, lines 441-444]

"However, the global-mean STDDEV of 500 hPa geopotential height for the AMSU-A run is reduced to about 42 m as compared with the STDDEV (about 49 m) for the CNTL run, meaning that the 500 hPa geopotential height predictions are improved by assimilating the AMSU-A observations (Table 2). In particular, the error is significantly reduced over the Northern Hemisphere."

[New]

However, the global-mean STDDEV of 500 hPa geopotential height for the AMSU-A run is reduced to about 42 m as compared with the STDDEV (about 49 m) for the CNTL run, meaning that the 500 hPa geopotential height predictions are improved by assimilating the AMSU-A observations (Table 2). In particular, the error is largely reduced over the Northern Hemisphere. That is, the analysis impact is more significant in the Northern Hemisphere. It is inconsistent with the consensus that the assimilation impact of satellite observations is larger in the Southern Hemisphere, where the conventional data are sparse (Terasaki and Miyoshi, 2017; Yamazaki et al., 2023).

Bormann, N., Kobayashi, S., Matricardi, M., McNally, A., Krzeminski, B., Thépaut, J.-N., and Bauer, P.: Recent developments in the use of ATOVS data at ECMWF. In Proceedings of the 16th International TOVS Study Conference, Angra dos Reis, Brazil. CIMSS, Univ. Wisonsin: Madison, USA, 2008.

English, S., Renshaw, R., Dibben, P., Smith, A., Rayer, P., Poulsen, C., Saunders, F., and Eyre, J.: A comparison of the impact of TOVS arid ATOVS satellite sounding data on the accuracy of numerical weather forecasts, Q. J. Roy. Meteor. Soc., 126, 2911–2931, 2000.

Terasaki, K. and Miyoshi, T.: Assimilating AMSU-A Radiances with the NICAM-LETKF, J. Meteorol. Soc. Jpn., 95, 433–446, https://doi.org/10.2151/jmsj.2017-028, 2017.

Yamazaki, A., Terasaki, K., Miyoshi, T., and Noguchi, S.: Estimation of AMSU-A radiance observation impacts in an LETKF-based atmospheric global data assimilation system: Comparison with EFSO and observing system experiments, Weather and Forecasting, 38, doi:10.1175/WAF-D-22-0159.1 2023.

*You abbreviate the standard deviation of the first-guess departure as STDDEV. I find this abbreviation very misleading and think that RMSE (or similar) would fit better, since STDDEV refers to standard-deviation and the standard-deviation of the forecast is the spread. So when I read your paper the first time, I thought that you show the reduction in spread. But with your definition STDDEV is the RMS difference (error) of observation minus background. Replacing the abbreviation STDDEV with RMSE (or similar) would be more clear.*

[Reply]

Thank you for your comment. Root-mean-square error (RMSE) and standard deviation (STDDEV) have similar formulas. However, two values are not the same. The STDDEV is used to measure the spread of data from the mean value. In contrast, the RMSE represents the spread between different two data (i.e., observation and background in this study). Thus, the RMSE includes the mean bias between the two data as well as the spread. In section 7.1, the STDDEV of first-guess departure was computed in order to remove the mean bias and only measure the background (i.e., 6-h forecast) error spread against the observation, as follows:

$$STDDEV_{OmB} = \sqrt{\frac{1}{N}\sum_{i=1}^{N}(OmB_i - \overline{OmB})^2}$$

$$RMSE_{OmB} = \sqrt{\frac{1}{N}\sum_{i=1}^{N}(OmB_i)^2}$$

Thus, it is better to keep the abbreviation of standard deviation to correctly describe the error values in section 7.1.

*L516: "in the tropics, the analysis impact is relatively small due to small model errors". Option A: I guess you mean "smaller background error" (see minor comment L462). If you refer to the RMSE being smaller in the tropics, then I suggest a change of "model errors" to e.g. "background errors" or "first-guess departures". Option B: If you really want to say "model error" then I don't see from what result you conclude this. Please provide reasoning, evidence or rephrase (e.g. "presumably due to") or tell the reader where it can be seen from (e.g. figure x). Why is the numerical model better in the tropics than at 60-90° S. I would rather guess that you get more analysis impact due to larger first-guess departures (option A) due to baroclinic waves?*

[Reply]
First, it seems to make you confused to mention the word "model error" in the summary. Considering that the background is also a short-term forecast (6-h) derived at the previous analysis cycle, it is better to replace "the model errors" with "forecast errors" as follows:
[Old, line 516]
"However, in the tropics, the analysis impact is relatively small due to the small model errors."
[New]
"However, in the tropics, the analysis impact is relatively small due to the small forecast errors."

[Reply]
This conclusion comes from Figure 8 and Figure 10. As shown in Figure 8, the analysis impact in the tropics is relatively smaller than that over the extratropics. It is also noted that the analysis error (about 14 m) is also quite small in the tropics as compared with the analysis errors in the extratropics (i.e., about 49 m and 63 m in the Northern Hemisphere and Southern Hemisphere, respectively). As the forecast evolves from the analysis over time, a large analysis error leads to a large forecast error and vice versa. Considering that the model background error covariance matrix is derived from the short-term ensemble forecasts, we expect that the background error covariance is also small in the tropics where the analysis error is small. Thus, the AMSU-A radiances are conservatively assimilated in the tropics as compared with the extratropics where the analysis errors are large. In addition, following Judt (2020), it was demonstrated that the predictability in the tropics is longer than in the middle-latitudes and polar regions because the equatorial waves are described well in the numerical model compared to baroclinic disturbances in the middle-latitude regions. We revised some sentences as follows:
[Old, lines 445-449]
"In the tropics, the analysis error (about 14 m) is quite small for the CNTL run, as compared with the large errors of about 48 m and 63 m in the Northern Hemisphere and Southern Hemisphere, respectively.

The small STDDEV over the tropics in the CNTL run (shown in Fig. 10a) suggests that the assimilation of the conventional data has brought the model ensembles into an agreement with the AMSU-A observations, so less improvement is there compared to the extratropics."

[New]

"In the tropics, the analysis error (about 14 m) is quite small for the CNTL run, as compared with the large errors of about 48 m and 63 m in the Northern Hemisphere and Southern Hemisphere, respectively. Following Judt (2020), it was demonstrated that the tropical atmosphere has longer predictability than the extratropical atmosphere. Thus, the AMSU-A observations are conservatively assimilated in the tropics due to the small forecast errors, leading to less analysis impact."

Judt, F.: Atmospheric predictability of the tropics, middle latitudes, and polar regions explored through global storm-resolving simulations, J. Atmos. Sci., 77, 257–276, https://doi.org/10.1175/JAS-D-19-0116.1, 2020.

*L515: You say that analysis errors in T, U, V were significantly reduced. You probably refer to figure 11, which shows "Normalized difference of the standard deviation". Does it show the standard deviation of the forecast (i.e. ensemble spread?) or the standard deviation of the first-guess departure (i.e. the root-mean-square error of the forecast – observation)? Please be specific. Option A: If this is the ensemble spread, then it is not the error of the forecast, making your statement "analysis errors were reduced" to be wrong. You could say that "following the reduced ensemble spread, we expect a significant error reduction due to AMSU-A". To measure the error of the forecast directly, it is necessary to compare to observations or independent analyses. Option B: If it is the RMS of forecast-observation, then replace "standard-deviation" with "RMSE of ... forecasts" or similar.*

[Reply]

To assess the analysis impact of AMSU-A observations, in this study, the "ensemble-mean" analysis fields are verified against the ERA5 reanalysis. Using the departures between the DART ensemble-mean analysis and the ERA5 for four primary atmospheric variables (i.e., 500 hPa geopotential height, temperature, zonal wind, and meridional wind), two errors (i.e., bias and standard deviation) are computed. Thus, the bias and standard deviation of 500 hPa geopotential height are shown in Figure 8. And the normalized difference of standard deviation of other variables between the experimental run and control run is plotted in Figure 11. To prevent confusion, we revised some sentences as follow:

[Old, lines 404-408]

"For four primary atmospheric parameters (i.e., 500 hPa geopotential height, temperature, zonal wind, and meridional wind), the analysis errors are computed. In particular, the skill score of 500 hPa geopotential height is widely used as one of the key indicators to assess the overall performance of the model-derived output, because large-scale atmospheric motion in the middle troposphere (500 hPa) is closely linked with lower-level atmospheric motion."

[New]

"For four primary atmospheric parameters (i.e., 500 hPa geopotential height, temperature, zonal wind, and meridional wind), the departures between the DART ensemble-mean analysis and the ERA5 are computed. Then bias and standard deviation are derived from the long-term departures. In particular, the error of 500 hPa geopotential height is widely used to assess the overall performance of the model-derived output, because large-scale atmospheric motion in the middle troposphere (500 hPa) is closely linked with lower-level atmospheric motion."

*L518: "the number of assimilated AMSU-A data is small " ... "because the AMSU-A data are not*

*assimilated in the harsh condition of high latitude regions" Instead of saying "due to harsh conditions", please provide a more scientific reason why observations are not assimilated. With information from L466, I would suggest to write e.g. "we rejected observations at latitudes >60° S because these observations degraded the analysis" or e.g. "due to detrimental effects of clouds and sea-ice", if this is the case.*

[Reply]
Thank you for your comment. To make this sentence clear, we revised this sentence as follows:
[Old, lines 516-519]
"Compared with the Northern Hemisphere, the number of assimilated AMSU-A data is small over the Southern Hemisphere, because the AMSU-A data are not assimilated in the harsh condition of high latitude regions (> 60°S) during the Southern Hemisphere winter season, resulting in a relatively small analysis impact over the Southern Hemisphere."
[New]
"Compared with the Northern Hemisphere, less analysis impact in the Southern Hemisphere seems to be due to the reduction in the number of assimilated AMSU-A observations. The AMSU-A observations are rejected in the high latitude regions (> 60°S) during the Southern Hemisphere winter season, because assimilating these observations worsens the analysis quality."

*I could not open your EXP_model_outputs.egg and CNTL_model_outputs.egg in the zenodo repository. Could you please indicate how this filetype can be read?*

[Reply]
I am sorry for making you inconvenient. As the model outputs were compressed in the rare file format "egg" which can be only decompressed with the ALZip software, it may not be easy to use this software. Thus, we re-uploaded these output files in the "tar" and "gzip" formats, the universal formats to compress/decompress the large-size files. Please, access this URL (https://doi.org/10.5281/zenodo.7983459) and download these model outputs in the Zenodo repository.

**Minor comments**

*L17: " is obtained by assimilating the AMSU-A observations on top of the DART data assimilation system that already makes use of the conventional measurements" DART itself is a software and users decide on which observations to assimilate. Do you refer to a specific DART system which makes use of conventional observations? If it is your own system, then I suggest to remove "the DART assimilation system that makes use of" giving you: "… assimilating AMSU-A observations on top of conventional measurements" to avoid confusion*

[Reply]
In this study, we used the version 9.11.13 of DART which is accessible via the DART data repository (https://github.com/NCAR/DART). In this version of DART, many types of conventional observations are able to be assimilated using the data-specific modules to convert the specific type of observation into the input format specified for the DART data assimilation system. In this study, we also used the original modules to assimilate the conventional observations without any modification in the DART package. To make this point clear, we revised this sentence as follows:
[Old, lines 52-53]
"In addition, well-defined modules are included to make various types of observations available in the

DART data assimilation process."
[New]
"In addition, well-defined modules are included to make various types of observations available in the DART data assimilation process. Thus, DART can assimilate many observation types (e.g., conventional and satellite-based wind)."

*L26: "With the advances in the observation/computation technique and the improved data assimilation methodology, the quality of the initial condition significantly increases" I suggest to rephrase, because it is not clear to me what the advance in observation /computation technique you refer to. Maybe you refer to increased amount of observations or satellite observations? What computation technique do you refer to? Parametrizations? An improved dynamical core?*

[Reply]
Thank you for your comments. We clarified these sentences as follows:
[Old, lines 26-27]
"With the advances in the observation/computation technique and the improved data assimilation methodology, the quality of the initial condition significantly increases, which enhances the forecast skill."
[New]
"With the huge amount of satellite observations and advances in model configurations (e.g., horizontal/vertical resolution, dynamic core) and data assimilation, the quality of the initial condition significantly increases, which enhances the forecast skill."

*L38: "However, researchers, who are not affiliated with the operational NWP centers, are restricted from accessing these data assimilation systems, because these operational NWP systems should be securely managed to provide global weather forecasting to the forecasters and users on time." I agree that this is the case for many operational centers. However, I think you don't need to explain why centers don't share their system for research and I think it is not necessary for the reader. I suggest to remove or rephrase like "In our experience, researchers, who are not affiliated ..." or "Currently, ..." because this could change in the future. I also suggest you focus on the advantages of your setup e.g. by saying that your setup is freely available to everyone.*

[Reply]
I agree with your suggestion. It is better to remove this sentence to prevent confusion.
[Old, lines 37-43]
"Operational NWP centers have well-constructed assimilation systems to use diverse types of available observations with up-to-date data assimilation schemes. However, researchers, who are not affiliated with the operational NWP centers, are restricted from accessing these data assimilation systems, because these operational NWP systems should be securely managed to provide global weather forecasting to the forecasters and users on time. In addition, as most operational global NWP systems are installed in high-performing computation systems due to the huge computation resources required, it is practically impossible to handle the operational NWP system under the computation environment in which sufficient computation resources are not provided."
[New]
"Operational NWP centers have well-constructed assimilation systems to use diverse types of available observations with up-to-date data assimilation schemes. However, as most operational global NWP systems require huge computation resources, it is practically impossible for researchers to recreate those

systems outside of the NWP centers."

*L63-67: It seems you use Zhou et al. 2022 as a reason ("thus") for why there is interest to assimilate satellite-observed radiances. Zhou et al. 2022 is an example for radiance assimilation using the same software (DART). I suggest rephrasing.*

[Reply]
I partly agree with your interpretation of this paragraph. From our point of view, the aim of introducing Zhou's paper in this paragraph is to highlight that the observed radiances of satellites are first assimilated into the DART system, compared with Zhou's paper in which the simulated radiances, not observed, were assimilated into the DART system. Of course, we also considered mentioning Zhou's paper in the previous paragraph that describes some previous DART assimilation studies in which other types of observations (i.e., GPSRO and wind profilers) were assimilated. But, it was contextually inappropriate to mention Zhou's paper in the previous paragraph, because the satellite-related contents should be described in the next paragraph including the research motivation. For this reason, we decided to introduce Zhou's paper in this paragraph to highlight the originality of our study. We modified this sentence as follow:
[Old, lines 65-67]
"Thus, it is of interest to assimilate the satellite-observed radiances using the DART data assimilation system, in order to know how the analysis derived from DART is affected by satellite observations."
[New]
"Considering that, it is interesting to assimilate the satellite-observed radiances using the DART data assimilation system to know how the analysis derived from DART is affected by real satellite observations."

*L119: I suggest to provide a direct link to the dataset for easier replication. I assume it is https://rda.ucar.edu/datasets/ds337.0/ ?*

[Reply]
Thank you for your comment. We revised this sentence as follow:
[Old, lines 117-119]
"The baseline observation data are obtained from the National Centers for Environmental Prediction (NCEP) Automated Data Processing (ADP) global upper air and surface weather observations that are available from the NCAR Research Data Archive (NCAR RDA) (https://rda.ucar.edu/)."
[New]
"The baseline observation data are obtained from the National Centers for Environmental Prediction (NCEP) Automated Data Processing (ADP) global upper air and surface weather observations that are available from the NCAR Research Data Archive (NCAR RDA) (https://rda.ucar.edu/datasets/ds337.0/)."

*L150: Is this "gross quality control" the same as the outlier_treshold option in DART? Please clarify.*

[Reply]
Yes, the "gross quality control" is the same as the outlier test in the quality control process embedded in the DART system. We replaced the "gross quality control" with the "outlier test" as follows:
[Old, lines 14-15]
"In the quality control, three sub-processes are included: gross quality control, channel selection, and

spatial thinning."
[New]
"In the quality control, three sub-processes are included: outlier test, channel selection, and spatial thinning."

[Old, lines 147-148]
"In the preprocessing, two main steps are included: quality control and bias correction. Quality control consists of three sub-processes: gross quality control, channel selection, and spatial thinning."
[New]
"In the preprocessing, two main steps are included: quality control and bias correction. Quality control consists of three sub-processes: outlier test, channel selection, and spatial thinning."

[Old, line 151]
", the AMSU-A observation is not assimilated (called gross quality control)."
[New]
", the AMSU-A observation is not assimilated (called outlier test)."

[Old, lines 505-507]
"To make the AMSU-A data available to be assimilated, preprocessing steps were developed, which include quality control (i.e., gross quality control, channel selection and spatial thinning) and bias correction (i.e., scan-bias correction and air-mass-bias correction)"
[New]
"To make the AMSU-A data available to be assimilated, preprocessing steps were developed, which include quality control (i.e., outlier test, channel selection and spatial thinning) and bias correction (i.e., scan-bias correction and air-mass-bias correction)"

*L219: "Biases change with time" I suggest a change to "biases depend on time-of-day and on the season". Change with time could mean that the biases are evolving, but the term "bias" usually means a constant (average), systematic error.*

[Reply]
I agree with your suggestion. We modified this sentence as follows:
[Old, lines 219-220]
"The biases tend to change with time (diurnal or seasonal), the scan position of the instrument, and air mass."
[New]
"The biases tend to depend on time-of-day and on the season as well as the instrument scan angle and air mass."

*Figure 8a and 9: Confidence intervals would be appreciated to see whether any differences are significant. I guess you could do bootstrapping or a test for the difference.*

[Reply]
We overlaid 99% confidence intervals in Figures 8 and 9 as follows:

[Figure]

Figure 8. (a) Mean bias and (b) standard deviation (STDDEV) of 500 hPa geopotential height over the global (grey), Northern Hemisphere (NH; blue), tropics (TR; green), and Southern Hemisphere (SH; red). Filled and hatched bars indicate the results for the control (CNTL) and experiment (AMSU-A) run, respectively. The 99% confidence intervals are indicated by the vertical black lines.

[Figure]

Figure 9. Mean bias of the first-guess departure for the radiosonde temperature measurements for the control (CNTL run; circle symbol and black line) and experiment (AMSU-A run; square symbol and red line) runs. Horizontal lines indicate 99% confidence intervals.

*Figure captions: instead of ";" please use ":", e.g. "(global: grey, tropics: green, ...)*

[Reply]

Thank you for comment. We revised the captions of figures in the manuscript.

*L352: What is a horizontal distribution? I suggest to remove "horizontal".*

[Reply]
We revised this sentence as follow:
[Old, lines 351-353]
"In addition to the reduction of localization half-width (compared to the default value of 0.15), the sampling error correction algorithm was applied, which uses pre-defined information about the horizontal distribution of the correlation between the model state variables and the observations as a function of ensemble size."
[New]
"In addition to the reduction of localization half-width (compared to the default value of 0.15), the sampling error correction algorithm was applied, which uses pre-defined information about the correlation between the model state variables and the observations as a function of ensemble size."

*L361: It could be interesting to add information on whether you used the inflation method with a gaussian distribution (option 2) or the inverse gamma (option 5) for the inflation value.*

[Reply]
We revised these sentences as follow:
[Old, lines 360-362]
"In DART, the ensemble spread varies spatiotemporally, as a function of the evolving observation network and the chosen inflation algorithm. These experiments use a spatiotemporally varying inflation algorithm."
[New]
"In DART, the ensemble spread varies spatiotemporally, as a function of the evolving observation network and the chosen inflation algorithm. These experiments use a spatiotemporally varying inflation algorithm with a gaussian distribution."

*L405: "Skill score of 500 hPa GPH". The term "skill score" is reserved for specific verification metrics. I suggest to remove "skill score" as it is not necessary or use " we verify the 500-hPa geopotential height using first-guess departure mean and standard deviation."*

[Reply]
Thank you for your comment. To make it clear, we revised this sentence as follow:
[Old, lines 405-408]
"For four primary atmospheric parameters (i.e., 500 hPa geopotential height, temperature, zonal wind, and meridional wind), the analysis errors are computed. In particular, the skill score of 500 hPa geopotential height is widely used as one of the key indicators to assess the overall performance of the model-derived output, because large-scale atmospheric motion in the middle troposphere (500 hPa) is closely linked with lower-level atmospheric motion."
[New]
"For four primary atmospheric parameters (i.e., 500 hPa geopotential height, temperature, zonal wind, and meridional wind), the departures between the DART ensemble-mean analysis and the ERA5 are computed. Then bias and standard deviation are derived from the long-term departures. In particular, the error of 500 hPa geopotential height is widely used to assess the overall performance of the modelderived output, because large-scale atmospheric motion in the middle troposphere (500 hPa) is closely linked with lower-level atmospheric motion."

*L459-460: "if both regions are extracted … the assimilation impact is comparable" means just that if no observations are assimilated, we would not get an analysis impact. I would suggest to write instead what would happen if it would not be August/September, but February/March? Would we have a comparable/larger/smaller analysis impact compared to the analysis impact in the Northern Hemisphere in August?*

[Reply]
This sentence seems to make us misunderstand. As shown in Figure 10, the analysis errors for the AMSU-A run are significantly reduced in the mid-latitude region (i.e., 30°S-60°S and 30°N-60°N) in both hemispheres, compared with the control run. However, in the high-latitude region (i.e., 60°S-90°S and 60°N-90°N), the analysis impact of AMSU-A observations is only significant in the Northern Hemisphere, and neutral in the Southern Hemisphere. Thus, if the high latitude regions are not considered in the error computation, the analysis impact is still significant, but the gap in the analysis impact (shown in Table 2) between both hemispheres considerably decreases. To prevent confusion, we revised Table 2 and some sentences as follows:

[Old, line 431]
Table 2. Error statistics of 500 hPa geopotential height (m) for the control (CNTL run) and experiment (AMSU-A run) run. Better values are bolded.

| Trial Name | Bias | | | | STDDEV | | | |
|---|---|---|---|---|---|---|---|---|
| | Global | NH | TR | SH | Global | NH | TR | SH |
| CNTL | -18.70 | **-13.90** | -19.05 | -27.45 | 48.82 | 48.02 | 13.55 | 62.54 |
| AMSU-A | **-18.59** | -17.39 | **-17.73** | **-25.51** | **42.42** | **31.55** | **12.41** | **58.29** |

[New]
Table 2. Error statistics of 500 hPa geopotential height (m) for the control (CNTL run) and experiment (AMSU-A run) run. Better values are bolded. In parentheses, error statistics are shown over the mid-latitude region (30°S-60°S and 30°N-60°N) in the Northern and Southern Hemisphere.

| Trial Name | Bias | | | | STDDEV | | | |
|---|---|---|---|---|---|---|---|---|
| | Global | NH | TR | SH | Global | NH | TR | SH |
| CNTL | -18.70 | **-13.90** (-18.43) | -19.05 | -27.45 (-19.84) | 48.82 | 48.02 (26.71) | 13.55 | 62.54 (38.55) |
| AMSU-A | **-18.59** | -17.39 (-16.95) | **-17.73** | **-25.51** (-19.54) | **42.42** | **31.55** (20.24) | **12.41** | **58.29** (33.49) |

[Old, lines 459-460]
"Thus, if the high-latitude regions (i.e., 60°S-90°S and 60°N-90°N) are extracted in the error computation over both hemispheres, the assimilation impact is comparable (not shown)."
[New]
"Thus, if the high-latitude regions (i.e., 60°S-90°S and 60°N-90°N) are extracted in the error computation over both hemispheres, the analysis impact is still significant, but the difference in the

analysis impact between both hemispheres considerably decreases."

In addition, as you suggested, it is also interesting how the analysis impact pattern changes if the trial period is the winter season (January – February) in the Northern Hemisphere. Following Yamazaki et al. (2023), the analysis impact is quite similar for each winter season in the Northern Hemisphere (21 Dec 2018 – 31 Jan 2019) and Southern Hemisphere (23 August – 18 October 2019).

*L462: "as the conventional observations are quite sparse in the high latitude region, the model errors are relatively larger than the other latitudinal regions" I suggest to use "forecast error" instead of "model error" if figure 10 shows the RMSE of the forecast. Model error specificly states that the model is wrong. Forecast error is more general and includes that missing observations lead to larger analysis error which then grow into large forecast errors. I guess the latter is the case.*

[Reply]
As you mentioned, the analysis quality is less improved if a small number of observations are assimilated, leading to the degradation of the short-term forecast quality which is used as the background for the next assimilation cycle. This assimilation circulation is continuously accumulated in the high latitude region where the conventional observations are rare, resulting in the overall reduction of the model performance (i.e., forecast error). Thus, considering this context, it is suitable to use the "forecast error" instead of the "model error" in this sentence as follow:
[Old, lines 462-464]
"In particular, as the conventional observations are quite sparse in the high latitude region, the model errors are relatively larger than the other latitudinal regions (i.e., the tropics and mid-latitude region, shown in Fig. 10a)."
[New]
"In particular, as the conventional observations are quite sparse in the high latitude region, the forecast errors are relatively larger than the other latitudinal regions (i.e., the tropics and mid-latitude region, shown in Fig. 10a)."

*L466: From figure 2, I see that the rejected observations coincide with the presence of sea-ice? What do you think? Maybe you can comment on that?*

[Reply]
As mentioned in section 4.1., three tropospheric channels (channels 5–7) are not assimilated over the land and sea-ice, because the surface information (e.g., surface temperature and surface spectral emissivity) is uncertain. Thus, in the channel selection step, the cloud liquid water (CLW) and sea-ice index (SII) are retrieved using two retrieval algorithms suggested by Grody et al. (2001) and Grody et al. (1999), respectively. That is, if the CLW is larger than 0.2 mm or the SII is larger than 0.1, three tropospheric channels (channels 5–7) are rejected. For this reason, a large number of rejected AMSU-A observations (shown in Fig. 2c) are shown over sea-ice areas (shown in Fig. 2b) near the north and south poles. This point was described as follows:
[Old, lines 183-190]
"In this study, seven candidate AMSU-A channels (i.e., channels 5–11) are assimilated differently, depending on the surface type. Channels 5, 6, and 7 are the main tropospheric channels. Their weighting functions peak below 200 hPa, but also have a bit of sensitivity to the surface because of the broad vertical shape of the weighting functions. Thus, the quality of the analysis can be degraded by assimilating the three tropospheric channels over the land and sea-ice types whose surface information

(e.g., surface temperature and surface spectral emissivity) is uncertain. For this reason, AMSU-A channels 5–7 are not assimilated over the land and sea ice. To identify sea-ice area, the sea-ice index (SII) is retrieved from observations of AMSU-A channels 1 and 3 over the high latitude region (poleward of 50 degrees), using the retrieval algorithm suggested by Grody et al. (1999)."

*L467: You write that observations from latitudes >60° S were rejected, but figure 2 shows also observations from >60°S at 180° longitude. Did you really exclude all observation latitude > 60°S or is it related to something else? e.g. an outlier threshold, or due to mostly cloud-affected observations with large CLW, sea-ice?*

[Reply]
Figure 2 aims to show how many AMSU-A observations are rejected if the cloud and sea-ice detection algorithms are applied in the channel selection step. After the cloud and sea-ice detection, the AMSU-A observations in the high latitude region (> 60°S) are rejected to prevent the analysis degradation. That is, these two steps (i.e., cloud/sea ice detection and the high-latitude rejection) are independently/sequentially proceeded. Figure 2a and 2b is the results of cloud and sea-ice detection, respectively. Finally, Figure 3c shows the spatial distribution of AMSU-A observations passed or rejected by two detections.

*L474: It is not clear to me whether ERA5 or radiosondes were used to produce figure 11.*

[Reply]
In Figure 11, the standard deviations of temperature and two wind (i.e., zonal and meridional winds) are computed against the ERA5 reanalysis. We mentioned the error computation of four primary variables in section 7.2 as follows:
[Old, lines 400-405]
"To assess the impact of the AMSU-A observations on the analysis derived from the DART data assimilation system, the analysis errors are computed between the DART analysis and the European Centre for Medium-Range Weather Forecasts (ECMWF) reanalysis version 5 (ERA5) as the reference data. As the ERA5 is made through the assimilation of all available observation data in the ECMWF data assimilation system and provides consistent maps without spatial gaps, the ERA5 is employed to assess the model-derived output. For four primary atmospheric parameters (i.e., 500 hPa geopotential height, temperature, zonal wind, and meridional wind), the analysis errors are computed."
[New]
"To assess the impact of the AMSU-A observations on the analysis derived from the DART data assimilation system, the analysis errors are computed between the DART analysis and the European Centre for Medium-Range Weather Forecasts (ECMWF) reanalysis version 5 (ERA5) as the reference data. As the ERA5 is made through the assimilation of all available observation data in the ECMWF data assimilation system and provides consistent maps without spatial gaps, the ERA5 is employed to assess the model-derived output. For four primary atmospheric parameters (i.e., 500 hPa geopotential height, temperature, zonal wind, and meridional wind), the departures between the DART ensemble-mean analysis and the ERA5 are computed. Then bias and standard deviation are derived from the long-term departures."

*Table 2: Is it possible to indicate which values are statistically significant? Or are they all significant?*

[Reply]

Using the error values described in Table 2, Figures 8a and 8b are made. We found that all values are statistically significant. In the updated Figure 8, we overlaid 99% significance intervals as follows:

[Figure]

Figure 8. (a) Mean bias and (b) standard deviation (STDDEV) of 500 hPa geopotential height over the global (grey), Northern Hemisphere (NH; blue), tropics (TR; green), and Southern Hemisphere (SH; red). Filled and hatched bars indicate the results for the control (CNTL) and experiment (AMSU-A) run, respectively. The 99% confidence intervals are indicated by the vertical black lines.

*Code and data availability: What does a potential user of your pre-processing modules have to do in order to use your modules? Do they have to download your code from zenodo or will the modules be integrated into DART? Maybe you can provide a "recipe" of which steps are necessary?*

[Reply]
If a potential user would like to assimilate the AMSU-A radiances into the DART system downloaded from the DART data repository (https://github.com/NCAR/DART), the user has to download some codes in the dataset (named "data_processing_codes.zip") uploaded in the Zenodo repository (https://doi.org/10.5281/zenodo.7714755). It is because the preprocessing steps are developed in this study, which are not provided in the original DART package. And then, these codes are needed to be integrated into the DART system. As we also uploaded the DART package (name "DART.zip") which was used in this study, a potential user can construct the AMSU-A radiance data assimilation system by referring to the code structure in our DART package. Hopefully, in near future, the preprocessing modules will be integrated into DART through close collaboration with NCAR.

*Figure 8, 10, 11: I guess these figures use ERA5 data? Please add the data source for which reference data has been used in the caption e.g. ERA5 reanalysis or observations.*

[Reply]
Thank you for your comment. We revised these captions as follows:

[Old, lines 418-420]
"Figure 8. (a) Mean bias and (b) standard deviation (STDDEV) of 500 hPa geopotential height over the global (grey), Northern Hemisphere (NH; blue), tropics (TR; green), and Southern Hemisphere (SH; red). Filled and hatched bars indicate the results for the control (CNTL) and experiment (AMSU-A) run, respectively."
[New]
"Figure 8. (a) Mean bias and (b) standard deviation (STDDEV) of 500 hPa geopotential height over the

global (grey), Northern Hemisphere (NH; blue), tropics (TR; green), and Southern Hemisphere (SH; red), derived against the ERA5 reanalysis. Filled and hatched bars indicate the results for the control (CNTL) and experiment (AMSU-A) run, respectively."

[Old, lines 471-472]
"Spatial distribution of the standard deviation (STDDEV) of the 500 hPa geopotential height for the (a) control run (CNTL) and (b) experiment (AMSU-A) runs."
[New]
"Spatial distribution of the standard deviation (STDDEV) of the 500 hPa geopotential height for the (a) control run (CNTL) and (b) experiment (AMSU-A) runs, derived against the ERA5 reanalysis."

[Old, lines 500-502]
"Figure 11. Normalized difference of the standard deviation (STDDEV) of (a) temperature, (b) zonal wind, and (c) meridional wind between the experiment (AMSU-A) run and the control (CNTL) run. Colors indicate the latitude regions (global; grey, Northern Hemisphere; blue, tropics; green, and Southern Hemisphere; red). Horizontal bars indicate 99% confidence intervals."
[New]
"Figure 11. Normalized difference of the standard deviation (STDDEV) of (a) temperature, (b) zonal wind, and (c) meridional wind between the experiment (AMSU-A) run and the control (CNTL) run, derived against the ERA5 reanalysis. Hatched colors indicate the latitude regions (global; grey, Northern Hemisphere; blue, tropics; green, and Southern Hemisphere; red). Horizontal lines indicate 99% confidence intervals."

---

## Author Comment (AC2)

**Referee #2**

*This study is related to the data assimilation of AMSE-A microwave radiance data (and additional PrepBUFR data) by DART (configured with EAKF) coupled with CESM. The AMSU-A observations were bias-corrected, and the observation errors were estimated. After these preprocessing steps, the observations were assimilated, and the results were validated and discussed. The study provided some interesting results, and the discussions were thought-provoking. Nevertheless, I have some concerns which were summarized below.*

We sincerely appreciate your comments and concerns about this manuscript. Detailed responses to your comments are precisely described below. In this response letter, to separate the referee's comments from authors' responses, your original comments are presented in an Italic font with an underline.

*1. L215-216: Do you mean that extra experiments were performed to determine the optimal spatial thinning length? If so, could you please provide some details about how the experiments were designed in order to estimate the optimal thinning length?*

[Reply]
As you mentioned, extra assimilation experiments were conducted using the same DART assimilation system, in order to decide the suitable spatial thinning distance between different AMSU-A pixels. To make this point clear, we revised some sentences as follow:
[Old, lines 215-216]
"In this study, the AMSU-A observations are spatially thinned at an interval of about 290 km that was empirically estimated with multiple pre-trial runs."
[New]
"To choose the optimal spatial thinning distance, we performed four extra assimilation runs in which different spatial thinning distance (i.e., 96 km, 192 km, 288 km and 384 km) was applied. These distances are multiples of the AMSU-A FOV footprint size (~48 km in nadir). The thinning interval of 288 km resulted in the largest analysis impact, so that distance was used to thin the observations in this study."

*2. L221-223: I have two comments here.*
*1) Could you please provide some references to this method (i.e., to estimate the bias by averaging the departures between the observed and simulated radiances)? Perhaps the study by Scheck et al. (2018) should be cited (section 5, P677). Although Scheck et al. (2018) focus on the visible imagery, their method should be applicable to the microwave imagery.*
*[Scheck, L., Weissmann, M., and Bernhard, M.: Efficient Methods to Account for Cloud-Top Inclination and Cloud Overlap in Synthetic Visible Satellite Images, J. Atmos. Ocean. Tech., 35, 665-685, doi:10.1175/JTECH-D-17-0057.1, 2018].*

[Reply]
We agree with the fact that the reflectance bias was made between the observed reflectance of "visible" satellite instrument and the modelled reflectance from the numerical model output in Scheck et al. (2018). We added this paper in the reference list as follows:
[Old, lines 221-223]
"In general, the biases are estimated using the time averaged departures between the observed radiances and the simulated radiances from the spatiotemporally collocated model field (background), because of

the absence of reference data suitable to compare the satellite observations."
[New]
"In these experiments, the biases are estimated using the time averaged departures between the observed radiances and the simulated radiances from the spatiotemporally collocated model field (background), because of the absence of reference data suitable to compare the satellite observations (Scheck et al., 2018)."

Scheck, L., Weissmann, M., and Bernhard, M.: Efficient Methods to Account for Cloud-Top Inclination and Cloud Overlap in Synthetic Visible Satellite Images, J. Atmos. Ocean. Tech., 35, 665-685, doi:10.1175/JTECH-D-17-0057.1, 2018.

*2) In my understanding, the bias of the observation could be estimated by this method, i.e, averaging the departures between the observed radiances and simulated radiances, only when the simulated radiances are unbiased statically. If the simulated radiances are biased, they cannot represent the "truth" very well. The simulated radiances are strongly influences by the model output of the pre-trial run. As is mentioned in L422-428, the model output of the pre-trial run is biased. Therefore, it seems that the estimated bias contains the observation bias and some model bias. About this problem, could you please give some explanations here?*

[Reply]
I agree with your opinion. The bias correction method applied in this study basically uses the departures between the observed radiances and the simulated radiances from the spatiotemporally collocated model forecast field (i.e., 6-h forecast). As you mentioned, for this correction method to work correctly, the simulated radiances should be unbiased, meaning that the model field (6-h forecast) is also bias-free. However, as it is still uncertain if the model fields are unbiased, many efforts have been taken to use the well-calibrated in-situ observations as the reference data. Nevertheless, it is not easy to find the reference observations able to compare the satellite radiances. For example, it seems to be the best way to compute the bias between the satellite-observed microwave radiances and the simulated radiances from the collocated radiosonde measurements (i.e., temperature and water vapor profiles) using the radiative transfer model. In this case, the number of radiosonde measurements is not enough to compute statistically significant values for the short-term period. In addition, the simulated radiances could be biased because the radiative transfer model (RTM) still has uncertainty, which is used to simulate the radiances from the radiosonde measurements. The research community related to satellite data assimilation has attempted to separate the integrated bias into the instrument bias and the model bias, but this bias correction issue is still not solved. Currently, thus, the biases are practically estimated using the departure between the satellite observations and the simulated observations from the model fields. To describe this point, we revised some sentences as follow:
[Old, lines 221-224]
"In general, the biases are estimated using the time averaged departures between the observed radiances and the simulated radiances from the spatiotemporally collocated model field (background), because of the absence of reference data suitable to compare the satellite observations."
[New]
"In these experiments, the biases are estimated using the time averaged departures between the observed radiances and the simulated radiances from the spatiotemporally collocated model field (background), because of the absence of reference data suitable to compare the satellite observations (Scheck et al., 2018). The use of the simulated radiances from the model background (i.e., 6-h forecast) may be questionable because the model background could be biased. However, it is effectively impossible to

find sufficient reference observations for comparing with these satellite observations, so the biases are made using the departures between the observed radiances and the model simulated radiances."

*3. L244: I was confused about the formula (4). Why was the averaged residual scan bias obtained by removing the mean bias of two near nadir FOVs (15 and 16) from the bias of the departure for each FOV (1–30)? In other words, why the off-nadir bias was estimated by subtracting the near-nadir bias? Could you please give more details here?*

[Reply]
Thank you for your comment. As mentioned in the manuscript, the AMSU-A instrument is a cross-track microwave sounder that scans 30 FOVs per horizonal scan line with the scan angle between ±48.33°. Thus, the optical path length between the earth and the instrument varies depending on the scan angle, called the limb effect. Considering that the scan bias correction aims to only remove the biases caused by the incorrect limb effect in the radiative transfer modeling, the bias in the near nadir (15 and 16 scan position) is mainly due to the air-mass bias, not the scan bias because the scan angles are almost zero. For this reason, as shown in Eq. 4, the residual scan bias is computed by extracting the averaged bias of two near-nadir FOVs (15 and 16) from the bias of each FOV (1–30).
To make this point clear, we modified some sentences as follows:
[Old, lines 238-239]
"Second, the averaged residual scan bias is obtained by removing the mean bias of two near-nadir FOVs (15 and 16) from the bias of the departure for each FOV (1–30)."
[New]
"Second, as the scan bias derived from the departures between the observed radiances and forward-modeled radiances likely includes the air-mass bias, the averaged residual scan bias is obtained by removing the mean bias of two near-nadir FOVs (15 and 16) from the bias for each FOV (1–30)."

*4. L337: Are twenty ensemble members enough for data assimilation in a global model? Were all ensemble members configured with the same physics options but with different initial and boundary conditions?*

[Reply]
All ensemble members have the same model configurations (e.g., model dynamics and physics), but are initialized from different initial/boundary conditions (i.e., atmosphere, land, and sea ice components). However, for the ocean component, the same sea surface temperature (SST) dataset is used for all ensemble members.
And, as you mentioned, the assimilation techniques based on the ensemble Kalman filter (EnKF) cannot be free from the sampling error that is caused by the limited size of the ensemble members. It is a fact that the larger the number of ensemble members, the higher the statistical significance of the correlation derived from the ensemble spread. However, as the number of ensemble members strongly depends on the computation capacity available to run the ensemble-based numerical model, it is difficult to increase the number of ensemble members indefinitely, in particular, in the small research community in which the high-performance computing system (e.g., super computer) is not installed. Thus, to eliminate the unreliable correlation derived by insufficient ensemble members, the localization method is applied. In addition to the localization, the sampling error correction scheme is available in DART, which uses pre-defined information about the correlation between the model state variables and the observations as a function of ensemble size. And, there are some previous studies in which 20 ensemble members were applied in the EnKF technique to assimilate the satellite-based output. In Mizzi et al. (2016), the

retrieval data of the satellite-based atmospheric composition were assimilated using the EnKF technique with twenty ensemble members. In addition, Zhang et al. (2021) assimilated the carbon dioxide ($CO_2$) retrievals from the Orbiting Carbon Observatory 2 (OCO-2) satellite into DART where the size of ensemble member was set to be twenty.

Mizzi, A. P., Arellano Jr., A. F., Edwards, D. P., Anderson, J. L., and Pfister, G. G.: Assimilating compact phase space retrievals of atmospheric composition with WRF-Chem/DART: a regional chemical transport/ensemble Kalman filter data assimilation system, Geosci. Model Dev., 9, 965–978, https://doi.org/10.5194/gmd-9-965-2016, 2016.

Zhang, Q. W., Li, M. Q., Wei, C., Mizzi, A. P., Huang, Y. J., and Gu, Q. R.: Assimilation of OCO-2 retrievals with WRF-Chem/DART: A case study for the Midwestern United States, Atmos. 985 Environ., 246, 118106, https://doi.org/10.1016/j.atmosenv.2020.118106, 2021.

*5. L344: A half-width of 0.075 radians is equivalent of a localization distance of 955.65 km (2\*6371\*0.075) in the horizontal direction. Could you please explain why is a localization distance of 955.65 km was set?*

[Reply]
Similar to the determination of AMSU-A spatial thinning distance (in 1st comment), a localization half-width of 0.075 radiance in this study was also chosen through three trial experiments. In three experiments, three localization half-widths of 0.15 (a default value), 0.075, and 0.0375 were applied. Among them, the largest analysis improvement was shown when the localization half-with was set as 0.075. A localization distance of about 1000 km (0.075 radians) also corresponds to a typical horizontal length scale of atmospheric synoptic phenomena (e.g., cyclones). To make this point clear, we revised some sentences as follow:
[Old, lines 344-345]
"In this study, the horizontal/vertical localization half-width of 0.075 radians was employed to prevent the use of erroneous correlation."
[New]
To determine the localization half-width, three extra assimilation experiments were run with different half-widths (i.e., 0.15, 0.075, and 0.0375). As the largest analysis impact was made with the half-width of 0.075, the horizontal/vertical localization half-width of 0.075 radians was employed to prevent the use of erroneous correlation."

*6. Figure 6 and Figure 11: Since different variables could interact with each other, as is mentioned in L383, I think it would be interesting to see how the humidity-related variables were influenced by assimilating the AMSU-A observations.*

[Reply]
Thank you for your comment. As the number of radiosonde relative humidity is quite small in the conventional observations obtained from the NCEP ADP, it is difficult to diagnose the model first-guess departure of the humidity against the radiosonde humidity measurements. Instead, we compute the error of specific humidity (g/kg or kg/kg) in the analysis against the ERA5 reanalysis to assess the impact of assimilating the AMSU-A observations on the model humidity field. As shown in Fig. S1b, the positive analysis impact (about -5 %) is shown in the troposphere (1000 hPa – 200 hPa) over the Northern Hemisphere where the analysis impact on the primary variables (i.e., 500 hPa geopotential height, temperature, zonal wind, and meridional wind) is significant. In the tropics and Southern Hemisphere,

the analysis impact is slightly negative, but not statistically significant, except in the lower stratosphere (near 100 hPa) in which the large negative impact (about 18 % and 15 % in the tropics and Southern Hemisphere, respectively) is shown. However, as shown in Fig. S1a, the magnitude of the standard deviation, a denominator to compute the normalized difference of the standard deviation (Fig. S1b), is quite small in the lower stratosphere for the control run. Thus, the change of the standard deviation seems to be negligible in the lower stratosphere. We mention this point in the revised manuscript as follows:

[New]

"In the model humidity field, a positive analysis impact only occurs in the Northern Hemisphere (not shown), but is not as significant as the abovementioned parameters (i.e., 500 hPa geopotential height, temperature, and winds). As a further study, we plan to assimilate the Microwave Humidity Sounder (MHS) providing information on the vertical structure of humidity so that the initial condition of model humidity is improved."

[Figure]

Figure S1. (a) The standard deviation of specific humidity (g/kg) for the control (CNTL) run and (b) normalized difference of the standard deviation of specific humidity between the experiment (AMSU-A) run and the control (CNTL). The 99% confidence intervals are indicated by the horizontal black lines.

*7. The assimilation would generate positive impact on the model variables at the analysis time. After that, the positive impact could diminish quickly with model integration since the balance law between different variables was not respected during the data assimilation process. It would be really helpful to see how the errors (biases, STDDEVs) for both the analysis and forecasting fields vary with time.*

[Reply]

In general, the forecasts are generated from the analysis field derived at each analysis cycle in the data

assimilation system comprising the forecast system and the data assimilation system. In the current version of the CESM/DART data assimilation system, long-range forecasts are not generated yet. However, the short-range forecast (i.e., 6-h forecast) is used as the background at each analysis cycle in the DART assimilation system, which is archived. As shown in Fig. S2a, the global-mean bias largely fluctuates during the trial period (11 August – 30 September 2014), but is negative for two runs overall. The STDDEV values fluctuate between 30 m and 70 m, but the AMSU-A run has an overall small STDDEV compared with the CNTL run, except for the period from 1 September to 9 September 2014 (Fig. S2b). As the DART reanalysis was used as the starting initial condition on 0000UTC 11 August 2014 for both runs, the STDDEV rapidly increases for the spin-up period (0000UTC 11 August to 1800UTC 24 August 2014). This pattern is also shown for the short-term forecast (6-h forecast). After the spin-up period (two weeks), all model variables are adjusted to be more consistent with the observations, which presumably represent a nearly balanced atmosphere, so at least some of the balance is preserved. In addition, as shown in Fig. S2, the analysis increments, which are the changes to model variables caused by the assimilation, are small, so any imbalance will be small.

[Figure]

Figure S2. Time series of (a) the bias and (b) standard deviation of 500 hPa geopotential height for the control (CNTL; black) run and experiment (AMSU-A; red) run during the trial period from 11 August to 30 September 2014). The solid and dashed lines indicate the errors (i.e., bias and standard deviation) of analysis and 6-h forecast (background), respectively.

---

## Author Comment (AC3)

**Referee #3**

*Review of "Assimilation of the AMSU-A radiances using the CESM (v2.1.0) and the DART (v9.11.13)/RTTOV (v12.3)"*

*This manuscript describes the efforts on assimilation AMSU-A radiances data in the DART system which is coupled with RTTOV123 and CESM. The procedures for quality control, spatial thinning, bias correction and observational errors calculation are developed. The positive impacts are found in the analysis of the primary atmosphere parameters. The paper is well-organized. It appears to be logically set out and the standard of English is acceptable. I recommend this paper for acceptance in the journal with several revisions.*

We sincerely appreciate your comments and concerns about this manuscript. Detailed responses to your comments are described below. In this response letter, to distinguish between the referee's comments and authors' responses, your original comments are presented in an Italic font with an underline.

**Major comments:**

*1, Line 150: "the square root of the sum of the observation error variance and the prior background error variance". How about adding a table to list the threshold of each channel?*

[Reply]
Thank you for your comment. As mentioned in the manuscript, the threshold value is defined by the square root of the sum of the observation error variance and the prior background error variance. Even though the observation error variance is pre-defined depending on the channel at each satellite platform (shown in Fig. 5), the prior background error variance spatially/temporally varies at each assimilation cycle. That is, the background error variance is derived using the 6-hour forecasts for 20 ensemble members at each analysis cycle. Thus, it is difficult to specify the threshold values employed to filter out the AMSU-A observations whose first-guess departure is large. To make this point clear, we revised this sentence as follows:
[Old, lines 148-151]
"If the difference between the observed AMSU-A brightness temperature and the forward-modeled brightness temperature derived from the model background (6-h forecast) is larger than three times the square root of the sum of the observation error variance and the prior background error variance, the AMSU-A observation is not assimilated (called gross quality control)."
[New]
If the difference between the observed AMSU-A brightness temperature and the forward-modeled brightness temperature derived from the model background (6-h forecast) is larger than three times the square root of the sum of the observation error variance and the prior background error variance, the AMSU-A observation is not assimilated (called outlier test). As the prior background error variance is based on the ensemble spread, the larger the ensemble spread of the 6-h forecast, the more the AMSU-A observations are assimilated."

*3, Line 4.2: How was the thinning interval "290 km" determined?*

[Reply]
The spatial thinning distance was empirically determined through the extra trial runs. To determine the

optimal thinning distance, we conducted four trial runs in which different thinning distances (i.e., 96 km, 192 km, 288 km, and 384 km) were applied. Among them, the largest analysis benefit was obtained when a thinning distance of 288 km was used. We modified some sentences as follows:

[Old, lines 215-216]

"In this study, the AMSU-A observations are spatially thinned at an interval of about 290 km that was empirically estimated with multiple pre-trial runs."

[Old]

"To choose the optimal spatial thinning distance, we performed four extra assimilation runs in which different spatial thinning distances (i.e., 96 km, 192 km, 288 km and 384 km) were applied. These distances are multiples of the AMSU-A FOV footprint size (~48 km in nadir). The thinning interval of 288 km resulted in the largest analysis impact, so that distance was used to thin the observations in this study."

*4, Line 239-242: The AMSU-A data with large latitude (>60°S) is excluded in this paper (Line 204), because their impacts are not ideal. Figure 3(b) shows that, "the residual san biases have different patterns depending on the latitude and for AMSU-A channel 6", and the scan biases are rather large on the latitude band 50S-60S, which is near the >60°S region. It can be expected that the scan biases on the latitude band 60S-90S should be larger. Is it the case? And is it the possible reason for the negative impacts of AMSU-A data >60°S?*

[Reply]

Thank you for your comment. As the residual scan biases were estimated using the assimilation outputs (e.g., assimilated observed radiances and background radiances) derived from two-week cycle run, the scan bias patterns seem to be representative of the trial period. And, as you expected, it was a fact that the residual scan bias was large on the high-latitude region (>60°S) in the Southern Hemisphere, as compared with the biases in other latitude regions.

We tried to know why the analysis quality was degraded when the AMSU-A observations in the high-latitude region (>60°S) were assimilated, but it remains unclear. One of the potential reasons is that the bias correction does not work correctly in the high-latitude region, in particular, under extremely cold weather conditions. As mentioned in the manuscript, the scan bias correction considers the characteristics of bias depending on the latitude band (shown in Fig. 3b), but the air-mass bias correction only uses the global-mean bias coefficients. Thus, the global-mean air-mass bias correction coefficients are likely not to work correctly in the high-latitude regions where different model bias pattern occurs locally, in particular, under the extreme weather condition. In fact, the trial experiments in this study were conducted for the summer season (11 August – 30 September 2014) in the Northern Hemisphere. In other words, the high-latitude region (>60°S) including the Antarctic continent was under extremely cold conditions. Thus, we guess that the analysis degradation occurs by applying the global-mean air-mass bias correction that does not consider the bias patterns in the high-latitude region (>60°S). This issue will be handled in the future study.

*5, Figure 4: The impact of bias correction on CH10 and CH11 seems not ideal. Especially, the histogram of OMB of CH11 appears to be more "skewed" distribution after bias correction than that before bias correction. And an average deviation of approximately 0.2 K is remained after bias correction. Why the bias correction on CH11 does not perform well? How about adding a table to list the mean value and standard deviations of OMB in each channel before and after bias correction?*

[Reply]

First, as the number of O-B samples is relatively small for the AMSU-A channel 11 compared with other channels (shown in Fig. 4 in the manuscript), we update this figure by extending the data period (three weeks from 25 August to 14 September 2014). And, to check the performance of the bias correction process, we add the table in which the mean biases and standard deviations of the first-guess departures are described before and after the bias correction process. For all candidate channels (AMSU-A channels 5–11 for MetOp-B satellite), the mean biases are close to zero, and the standard deviations slightly decrease when the bias correction scheme is applied for the AMSU-A radiances. However, as you mentioned, the distribution pattern of the O-B histogram of AMSU-A channel 11 is slightly skewed after the bias correction compared with other channels showing the Gaussian distribution (Fig. 4). It means that the air-mass bias correction process using the global-mean coefficients is not optimal for the AMSU-A channel 11. It is still unclear why the O-B histogram is skewed for AMSU-A channel 11. Thus, the further study is needed to enhance the performance of the bias correction process.

[Figure]

Figure 4. Histogram of the first-guess departures between the observations of the MetOp-B AMSU-A channels 5–11 and the corresponding model background (6-h forecast). Colors indicate the results before the bias correction (hatched blue) and after the bias correction (red), respectively.

Table 2. Mean biases and standard deviations of the first-guess departures (O-B) for MetOp-B AMSU-A channels before and after the bias correction.

| O-B | Bias correction | CH5 | CH6 | CH7 | CH8 | CH9 | CH10 | CH11 |
|---|---|---|---|---|---|---|---|---|
| Bias | X | 1.518 | 1.181 | 0.514 | 0.937 | 0.514 | 0.590 | 0.612 |
| | O | 0.0005 | 0.002 | 0.003 | 0.014 | 0.033 | 0.028 | 0.010 |
| STDDEV | X | 0.677 | 0.489 | 0.521 | 0.572 | 0.639 | 0.688 | 1.052 |
| | O | 0.627 | 0.482 | 0.494 | 0.554 | 0.580 | 0.642 | 0.966 |

*6, Figure 6 and Figure 11: The analysis results of temperature, zonal wind and meridional wind is given in detail, while the results of humidity are neither shown in the figures nor mentioned in the text. Although the channels on AMSU-A are mainly sensitive to the temperature, however, as mentioned in Line 382, "a change in one model parameter can change another model parameter in the assimilation process". In my experience, the assimilation of microwave temperature sounders will more or less bring some impacts on the humidity. I wonder the impacts on humidity analysis in this work.*

[Reply]

Thank you for your comment. As you mentioned, we computed the error of specific humidity (g/kg or kg/kg) in the analysis against the ERA5 reanalysis to assess the impact of assimilating the AMSU-A observations on the model humidity field. As shown in Fig. S1b, the positive analysis impact (about - 5 %) is shown in the troposphere (1000 hPa – 200 hPa) over the Northern Hemisphere where the analysis impact on the primary variables (i.e., 500 hPa geopotential height, temperature, zonal wind, and meridional wind) is significant. In the tropics and Southern Hemisphere, the analysis impact is slightly negative, but not statistically significant, except in the lower stratosphere (near 100 hPa) where the large negative impact (about 18 % and 15 % in the tropics and Southern Hemisphere, respectively) is shown. However, as shown in Fig. S1a, the magnitude of the standard deviation, a denominator to compute the normalized difference of the standard deviation (Fig. S1b), is quite small in the lower stratosphere for the control run. Thus, the change of the standard deviation seems to be negligible in the lower stratosphere. We mention this point in the revised manuscript as follows:

[New]

"In the model humidity field, a positive analysis impact only occurs in the Northern Hemisphere (not shown), but is not as significant as the abovementioned parameters (i.e., 500 hPa geopotential height, temperature, and winds). As a further study, we plan to assimilate the Microwave Humidity Sounder (MHS) providing information on the vertical structure of humidity so that the initial condition of model humidity is improved."

[Figure]

Figure S1. (a) The standard deviation of specific humidity (g/kg) for the control (CNTL) run and (b) normalized difference of the standard deviation of specific humidity between the experiment (AMSU-A) run and the control (CNTL). The 99% confidence intervals are indicated by the horizontal black lines.

*7, Figure 11: For both geopotential height, temperature and wind, the impacts of assimilating AMSU-A data in the Northern Hemisphere are better than that in the Southern Hemisphere (Line 444-445, 483-485, 491-493), and these are attributed to the lack of data assimilation in the region >60°S (494-497). However, the observational data from several channels of AMSU-A are also rejected over land (Table 1), which are mostly distributed in the Northern Hemisphere. In another word, the total amount of AMSU-A data assimilated in the Southern Hemisphere should be still more than those in the Northern Hemisphere. In general, the assimilation of satellite data brings more benefits to the analysis and forecasting of Southern Hemisphere, because of the larger ratio of ocean area and the lack of conventional observations. How to understand the difference between the results in this paper and our expectation?*

[Reply]
I agree with your opinion. It is well known that the satellite assimilation impact is generally larger in the Southern Hemisphere than its impact in the Northern Hemisphere where the conventional observations are plentiful. In addition, many observations of three tropospheric channels (channels 5–7) are only assimilated over the ocean, which covers more than half of the surface in the Southern Hemisphere. However, in this study, the analysis error reduction is similar in both hemispheres if the high latitude regions (>60°N and >60°S) are not considered in the error computation (see the parentheses in Table 2). It is still questionable why the satellite impact on the analysis is more significant in the Northern Hemisphere. One of the potential reasons is the absence of anchor observations (e.g., radiosonde and GPS-RO) in the Southern Hemisphere, which prevent a drift of the numerical model to its own climatology and biases (Cucurull et al., 2014). In the Southern Hemisphere, where radiosonde observations are rare, the GPS-RO observations are mainly used as the anchor observations. However, in this study, the GPS-RO observations were not assimilated in the trial experiment runs. We also attempted to include the GPS-RO observations in the baseline observations that were assimilated in the control run, because the DART package has the modules for these data. However, it was found that the GPS-RO observations did not provide the analysis benefits. We concluded that additional studies (e.g., error estimation and quality control) are needed to assimilate the GPS-RO observation, thus extracted these data from the baseline observations. In the future study, we will handle this issue.

[Old, line 431]
Table 2. Error statistics of 500 hPa geopotential height (m) for the control (CNTL run) and experiment (AMSU-A run) run. Better values are bolded.

| Trial Name | Bias | | | | STDDEV | | | |
|---|---|---|---|---|---|---|---|---|
| | Global | NH | TR | SH | Global | NH | TR | SH |
| CNTL | -18.70 | **-13.90** | -19.05 | -27.45 | 48.82 | 48.02 | 13.55 | 62.54 |
| AMSU-A | **-18.59** | -17.39 | **-17.73** | **-25.51** | **42.42** | **31.55** | **12.41** | **58.29** |

[New]
Table 2. Error statistics of 500 hPa geopotential height (m) for the control (CNTL run) and experiment (AMSU-A run) run. Better values are bolded. In parentheses, error statistics are shown over the mid-latitude region (30°S-60°S and 30°N-60°N) in the Northern and Southern Hemisphere.

| Trial | Bias | STDDEV |
|---|---|---|

| Name | Global | NH | TR | SH | Global | NH | TR | SH |
|:---:|:---:|:---:|:---:|:---:|:---:|:---:|:---:|:---:|
| CNTL | -18.70 | **-13.90** (-18.43) | -19.05 | -27.45 (-19.84) | 48.82 | 48.02 (26.71) | 13.55 | 62.54 (38.55) |
| AMSU-A | **-18.59** | -17.39 (-16.95) | **-17.73** | **-25.51** (-19.54) | 42.42 | **31.55** (20.24) | **12.41** | **58.29** (33.49) |

Cucurull, L., Anthes, R.A., Tsao, L.-L.: Radio Occultation Observations as Anchor Observations in Numerical Weather Prediction Models and Associated Reduction of Bias Corrections in Microwave and Infrared Satellite Observations, J. Atmos. Ocean. Tech., 31, 20–32, doi: 10.1175/JTECH-D-13-00059.1, 2014.

***Minor comments:***

*1, Figure 2 and Figure 6: the figures should be located behind the paragraph which first mentions it, i.e., behind Line 200 and 372, respectively (unless this manuscript is edited by LaTeX).*

[Reply]
Thank you for your comment. We will relocate these figures in the revised manuscript.

*2, Line 170-205: The authors describe the quality control procedures as three parts: gross quality control, channel selection, and spatial thinning (Line 15 and 506). However, generally speaking, the contents in Line 170-205 cannot be summarized by "channel selection", because these criterions are applied to the pixels instead of the whole channel. Besides, the spatial thinning is not belonged to quality control, because some pixels are rejected in this procedure not because their quality is not good. Thus, these paragraph should be reorganized. A simple consideration is to replace the title of section 4.1 by "quality control", and revise the corresponding statements in the abstract and conclusion.*

[Reply]
Thank you for your comment. We will reflect this point in the revised manuscript.

*3, Line 197: "Figures 2a and b" à "Figure 2a and b".*

[Reply]
We revise this sentence as follows:
[Old, line 197-198]
"As an example, Figures 2a and b present the spatial distribution of the CLW and the SII retrieved from AMSU-A on board NOAA-19 on 12 August 2014."
[New]
"As an example, Figure 2a and b present the spatial distribution of the CLW and the SII retrieved from AMSU-A on board NOAA-19 on 12 August 2014."

*4, Throughout the whole manuscript, sometimes "Figure" is worded, but sometimes "Fig." is worded. Please check the manuscript and unify it.*

[Reply]
Following the GMD policy, if the figure is mentioned at the beginning of the sentence, "Figure" is

worded. However, if the figure is referred to in the middle of the sentence, "Fig." is worded. For this reason, two words (i.e., "Figure" and "Fig.") are used in the manuscript depending on the location where the figure is mentioned in the sentence.

*5, Line 307: "In the pre-trial run, the instrument noise errors were initially used as the observation errors within DART." How long is the pre-trial run which is used for the statistics of observation errors?*

[Reply]
To estimate the observation error variance of AMSU-A channels onboard four satellite platforms, the pre-trial run was conducted for the period from 11 August to 30 September 2014. Except the spin-up period (11 August – 24 August 2014), the background innovations (O-B) and analysis innovations (O-A) were obtained from the output for the pre-trial run. And then, the observations error variances were computed using the estimation method suggested by Desroziers et al. (2005). To make this point clear, we revise this sentence as follow:
[Old, lines 305-307]
"To compute the observation error variances of AMSU-A channels on board four satellite platforms (i.e., Aqua, NOAA-19, MetOp-A, and MetOp-B), the background and analysis innovations were derived from the pre-trial run."
[New]
"To compute the observation error variances of AMSU-A channels on board four satellite platforms (i.e., Aqua, NOAA-19, MetOp-A, and MetOp-B), the background and analysis innovations were derived from the pre-trial run that was conducted from 25 August to 30 September 2014."

*6, Line 327: "CTRL" à "CNTL".*

[Reply]
Thank you for your comment. We revise this sentence as follow:
[Old, liens 325-327]
"the AMSU-A observations from four LEO satellite platforms (i.e., Aqua, NOAA-19, MetOp-A, and MetOp-B) were assimilated as well as the conventional data that were assimilated in the CTRL run."
[New]
"the AMSU-A observations from four LEO satellite platforms (i.e., Aqua, NOAA-19, MetOp-A, and MetOp-B) were assimilated as well as the conventional data that were assimilated in the CNTL run."

*7, Line 379: "Figs. 6b and c" à "Fig. 6b and c".*

[Reply]
Thank you for your comment. We revise this sentence as follow:
[Old, lines 378-379]
"In addition to the radiosonde temperature, the first-guess departure errors decrease for the two wind components (i.e., zonal and meridional winds) (Figs. 6b and c)."
[New]
"In addition to the radiosonde temperature, the first-guess departure errors decrease for the two wind components (i.e., zonal and meridional winds) (Fig. 6b and c)."

*8, If it is possible, all the figures are better to be parachromatism-friendly. The bars in Figure 4 and 11 are suggested to be shaded by different patterns, just like Figure 8. The symbol should be distinguished*

*not only by colors but also by shapes (squares, circles, triangles…) in Figure 5, 7, and 9.*

[Reply]

We update the figures as follows:

[Figure]

Figure 4. Histogram of the first-guess departures between the observations of the MetOp-B AMSU-A channels 5–11 and the corresponding model background (6-h forecast). Colors indicate the results before the bias correction (hatched blue) and after the bias correction (red), respectively.

[Figure]

Figure 5. Estimated observation errors (K) for AMSU-A channels on board Aqua (black; circle), NOAA-19 (red; square), MetOp-A (blue; diamond), and MetOp-B (green; triangle) satellite platforms. Black asterisks indicate the instrument noise errors for AMSU-A channels.

[Figure]

Figure 6. The standard deviation (STDDEV) of the first-guess departures for the radiosonde (a) temperature, (b) zonal wind, and (c) meridional wind for the control (CNTL run; circle symbol and black line) and experiment (AMSU-A run; square symbol and red line) runs. Solid and dashed lines indicate the STDDEV and the number (top axis) of radiosonde measurements assimilated, respectively. The 99% confidence intervals are indicated by the horizontal lines.

[Figure]

Figure 7. The standard deviations (STDDEVs) of the first-guess departure (unfilled symbols) and analysis departure (filled symbols) for AMSU-A channels on board Aqua (black; circle), NOAA-19 (red; square), MetOp-A (blue; diamond), and MetOp-B (green; triangle) satellites.

[Figure]

Figure 9. Mean bias of the first-guess departure for the radiosonde temperature measurements for the control (CNTL run; circle symbol and black line) and experiment (AMSU-A run; square symbol and red line) runs. Horizontal lines indicate 99% confidence intervals.

[Figure]

Figure 11. Normalized difference of the standard deviation (STDDEV) of (a) temperature, (b) zonal wind, and (c) meridional wind between the experiment (AMSU-A) run and the control (CNTL) run, derived against the ERA5 reanalysis. Hatched colors indicate the latitude regions (global; grey, Northern Hemisphere; blue, tropics; green, and Southern Hemisphere; red). Horizontal lines indicate 99% confidence intervals.

---

## Author Response (AR2)

Dear Topical editor,

Thank you for your comment. As Dr. Juan A. Añel pointed out, the data availability statement is updated in the revised manuscript as below. As well as the scripts used for the data analysis and visualization, the source codes of the models (i.e., DART and CESM) used in this study are uploaded at https://doi.org/10.5281/zenodo.7714755. In addition, the model outputs are saved at https://doi.org/10.5281/zenodo.7983459, including the DART-derived analysis and the model 6-h forecasts as the model backgrounds.

[New]
**Code and data availability.**
DART version 9.11.13 was obtained from https://github.com/NCAR/DART. CESM version 2.1.0 is released at https://github.com/ESCOMP/CESM/tree/release-cesm2.1.0. Atmospheric initial conditions and the baseline observations at the BUFR format were obtained from the NCAR RDA (https://rda.ucar.edu/datasets/ds337.0 or https://doi.org/10.5065/Z83F-N512). AMSU-A Level-1B version 5 data from the Aqua satellite, including the calibrated brightness temperatures, were downloaded from the NASA Goddard Earth Sciences Data and Information Services Center (https://www.earthdata.nasa.gov/eosdis/daacs/gesdisc). In addition, AMSU-A Level-1B from NOAA-19, MetOp-A, and MetOp-B satellites were downloaded from the atmosphere product section in the EUMETSAT product navigator (https://navigator.eumetsat.int). The ECMWF ERA5 hourly data on pressure levels were acquired from the Copernicus Climate Change Service (C3S) Climate Data Store (https://cds.climate.copernicus.eu/cdsapp#!/dataset/reanalysis-era5-pressure-levels). As well as the software codes, the model outputs are available at https://doi.org/10.5281/zenodo.7714755 and https://doi.org/10.5281/zenodo.7983459.

**Referee #1**

*Thank you very much for your substantial contribution. I like your study and how it is presented. Please clarify one issue: In figure 8, 10 and 11 you write "standard deviation (STDDEV) of the 500 hPa geopotential height" which could mean ensemble spread, but actually you mean "standard deviation of first-guess departures of the 500 hPa geopotential height". Please indicate in the caption somehow that it is the standard deviation of first-guess departures and \*not\* the standard deviation of the forecast. For example, write "standard deviation of first-guess departures".*

[Reply]
Thank you for your comment. In Figures 8, 10, and 11, the errors (i.e., mean bias and standard deviation) of four atmospheric variables (i.e., 500 hPa GPH, temperature, and zonal/meridional wind) were computed by comparing the DART-derived analysis (not first guess) with the ERA5 reanalysis as the reference. To clarify this point, we revised these captions as follows:

[Old]
"Figure 8. (a) Mean bias and (b) standard deviation (STDDEV) of 500 hPa geopotential height over the global (grey), Northern Hemisphere (NH: blue), tropics (TR: green), and Southern Hemisphere (SH: red), derived against the ERA5 reanalysis. Filled and hatched bars indicate the results for the control (CNTL) and experiment (AMSU-A) run, respectively. The 99% confidence intervals are indicated by the vertical black lines."
[New]
"Figure 8. (a) Mean bias and (b) standard deviation (STDDEV) of the analysis of 500 hPa geopotential height over the global (grey), Northern Hemisphere (NH: blue), tropics (TR: green), and Southern Hemisphere (SH: red), derived against the ERA5 reanalysis. Filled and hatched bars indicate the results for the control (CNTL) and experiment (AMSU-A) run, respectively. The 99% confidence intervals are indicated by the vertical black lines."

[Old]
"Figure 10. Spatial distribution of the standard deviation (STDDEV) of the 500 hPa geopotential height for the (a) control run (CNTL) and (b) experiment (AMSU-A) runs, derived against the ERA5 reanalysis."
[New]
"Figure 10. Spatial distribution of the standard deviation (STDDEV) of the analysis of 500 hPa geopotential height for the (a) control run (CNTL) and (b) experiment (AMSU-A) runs, derived against the ERA5 reanalysis."

[Old]
"Figure 11. Normalized difference of the standard deviation (STDDEV) of (a) temperature, (b) zonal wind, and (c) meridional wind between the experiment (AMSU-A) run and the control (CNTL) run, derived against the ERA5 reanalysis. Hatched colors indicate the latitude regions (global: grey, Northern Hemisphere: blue, tropics: green, and Southern Hemisphere: red). Horizontal lines indicate 99% confidence intervals."
[New]
"Figure 11. Normalized difference of the standard deviation (STDDEV) of the analysis of (a) temperature, (b) zonal wind, and (c) meridional wind between the experiment (AMSU-A) run and the control (CNTL) run, derived against the ERA5 reanalysis. Hatched colors indicate the latitude regions

(global: grey, Northern Hemisphere: blue, tropics: green, and Southern Hemisphere: red). Horizontal lines indicate 99% confidence intervals."

**Referee #2**

*Thank the authors to address my previous comments. The manuscript has been significantly improved. I have only one comment as below.*

*Several pre-trial runs were performed in the bias correction, error estimation, and etc. Could the authors please provide some details about the experiment designs and model settings of the pre-trial runs?*

[Reply]
Thank you for your comment. As you mentioned, many pre-trial runs were performed to estimate the bias correction coefficients (i.e., scan-bias correction coefficients), optimal spatial thinning, localization half-width, and the observation errors. As the pre-trial runs were run to determine the specific values of each assimilation factor, the detailed setup was different for each pre-trial run depending on which factor was determined. We revised some sentences as follow:

[Old, lines 221-222]
"To choose the optimal spatial thinning distance, we performed four extra assimilation runs in which different spatial thinning distance (i.e., 96 km, 192 km, 288 km and 384 km) was applied."
[New]
"To choose the optimal spatial thinning distance, we performed four extra assimilation runs in which different spatial thinning distance (i.e., 96 km, 192 km, 288 km and 384 km) was applied. Except for the spatial thinning distance, these pre-trial runs were set up with the same assimilation factors, i.e., the estimated bias correction coefficients (refer to section 4.3), the estimated observation errors (refer to section 5), and the localization half-width of 0.075 (refer to section 6)."

[Old, lines 252-254]
"First, the mean bias of the departure between the AMSU-A observed radiances and forward-modeled radiances for each FOV is made with the data assimilation results derived from the pre-trial run."
[New]
"First, the mean bias of the departure between the AMSU-A observed radiances and forward-modeled radiances for each FOV is made with the data assimilation results derived from the pre-trial run. The pre-trial run was set up with the spatial thinning of 96 km (refer to section 4.2) and the default localization half-width (0.15, refer to section 6). The instrument noise errors were used as the observation errors within DART."

[Old, lines 324-325]
"In the pre-trial run, the instrument noise errors were initially used as the observation errors within DART."
[New]
"In the pre-trial run, instrument noise errors were simply used as the observation errors. The pre-trial run was set up with the default localization half-width (0.15, refer to section 6), the spatial thinning of 96 km (refer to section 4.2), and the bias correction scheme (refer to section 4.3)."

[Old, lines 360-361]
"To determine the localization half-width, three extra assimilation experiments were run with different half-widths (i.e., 0.15, 0.075, and 0.0375)."
[New]

"To determine the localization half-width, three extra assimilation experiments were run with different half-widths (i.e., 0.15, 0.075, and 0.0375). Except for the localization half-width, the assimilation experiments were set up with the spatial thinning of 96 km (refer to section 4.2), the bias correction scheme (refer to section 4.3), and the estimated observation errors (refer to section 5)."